# Ancient mechanisms for the evolution of the bicoid homeodomain's function in fly development

Qinwen Liu[1†], Pinar Onal[2†], Rhea R Datta[2†‡], Julia M Rogers[3,4], Urs Schmidt-Ott[5], Martha L Bulyk[3,4,6], Stephen Small[2*], Joseph W Thornton[1,7*]

[1]Department of Ecology and Evolution, University of Chicago, Chicago, United States; [2]Department of Biology, New York University, New York, United States; [3]Committee on Higher Degrees in Biophysics, Harvard University, Cambridge, United States; [4]Division of Genetics, Department of Medicine, Brigham and Women's Hospital and Harvard Medical School, Boston, United States; [5]Department of Organismal Biology and Anatomy, University of Chicago, Chicago, United States; [6]Department of Pathology, Brigham and Women's Hospital and Harvard Medical School, Boston, United States; [7]Department of Human Genetics, University of Chicago, Chicago, United States

*For correspondence:
sjs1@nyu.edu (SS);
joet1@uchicago.edu (JWT)

[†]These authors contributed equally to this work

Present address: [‡]Biology Department, Hamilton College, Clinton, New York, United States

Competing interests: The authors declare that no competing interests exist.

**Abstract** The ancient mechanisms that caused developmental gene regulatory networks to diversify among distantly related taxa are not well understood. Here we use ancestral protein reconstruction, biochemical experiments, and developmental assays of transgenic animals carrying reconstructed ancestral genes to investigate how the transcription factor Bicoid (Bcd) evolved its central role in anterior-posterior patterning in flies. We show that most of Bcd's derived functions are attributable to evolutionary changes within its homeodomain (HD) during a phylogenetic interval >140 million years ago. A single substitution from this period (Q50K) accounts almost entirely for the evolution of Bcd's derived DNA specificity in vitro. In transgenic embryos expressing the reconstructed ancestral HD, however, Q50K confers activation of only a few of Bcd's transcriptional targets and yields a very partial rescue of anterior development. Adding a second historical substitution (M54R) confers regulation of additional Bcd targets and further rescues anterior development. These results indicate that two epistatically interacting mutations played a major role in the evolution of Bcd's controlling regulatory role in early development. They also show how ancestral sequence reconstruction can be combined with in vivo characterization of transgenic animals to illuminate the historical mechanisms of developmental evolution.
DOI: https://doi.org/10.7554/eLife.34594.001

## Introduction

Animal body plans are established during development by interacting networks of transcription factors and other molecules, which organize temporal and spatial cascades of gene expression that control cell division, differentiation, and migration (*Davidson and Peter, 2015*; *Gilbert, 2013*; *Stathopoulos and Levine, 2005*). How specific genetic changes during evolution modified these networks and the developmental processes they control is the fundamental question in modern studies of the evolution of development (evo-devo) (*Carroll, 2008*; *Davidson and Erwin, 2006*; *Levine and Tjian, 2003*; *Peter and Davidson, 2011*; *Rebeiz et al., 2015*).

Considerable progress has been made toward this goal, particularly among closely related species and populations. Comparative analyses have identified differences in gene expression that

correlate with phenotypic differences, and manipulative experiments have shown that altering expression of specific genes can disrupt or mimic specific forms of developmental diversity (*Angelini and Kaufman, 2005*; *Bhullar et al., 2015*; *Cretekos et al., 2008*; *Jarvis et al., 2012*; *Martin et al., 2016*; *Nakamura et al., 2016*; *Tarazona et al., 2016*; *Mazo-Vargas et al., 2017*; *Zhang et al., 2017*; *Roeske et al., 2018*). Transgenic experiments in which cis-regulatory regions are swapped between species or populations have identified polymorphisms that reproduce diversity in gene expression and developmental outcomes, such as pigmentation patterns in insects (*Gompel et al., 2005*; *Jeong et al., 2008*) and mice (*Manceau et al., 2011*), patterns of cuticular projections on the abdomens of flies (*Frankel et al., 2011*; *McGregor et al., 2007*), and the presence/absence of armor, spines, and other decorations among fish populations (*Chan et al., 2010*; *Colosimo et al., 1928*; *Miller et al., 2007*; *Shapiro et al., 2004*).

For deeper taxonomic divergences, it has been more difficult to establish the specific causes of evolutionary changes in gene regulation and development (*Rebeiz et al., 2015*). Distantly related taxa cannot be mated to each other, so genetic crosses cannot be used. Comparisons among species have demonstrated that candidate genes in key regulatory pathways differ in expression, and experimental manipulations have provided evidence that changes in expression of transcription factors in these pathways are involved in phenotypic differences (*Martin et al., 2016*; *Nakamura et al., 2016*; *Gehrke and Shubin, 2016*; *Leal and Cohn, 2016*; *Petit et al., 2017*). But the specific mutations and causal mechanisms that caused loss, gain, or modification of ancestral modes of regulation and developmental outcomes have not been identified, for several reasons. Distantly related genes (or regulatory elements) typically differ at many sequence sites – not only those that caused functional change but also those that diversified during subsequent periods of functional conservation and sequence drift – complicating efforts to identify the effect of specific substitutions (*Rebeiz et al., 2015*; *Hochberg and Thornton, 2017*). Epistasis among sites within loci represents a further challenge, because mutations introduced into present-day sequences often have effects different from those they had during historical evolution (*Bridgham et al., 2009*; *Natarajan et al., 2013*; *Ortlund et al., 2007*). In such cases, sequence differences that were causal in the past can be masked in horizontal swap experiments, leading to false negative inferences concerning the mutation's effects; alternatively, if a sequence state from one lineage is swapped into a present-day gene with which it is epistatically incompatible, that state may appear to have been the evolutionary cause of a null phenotype, a particular problem if the diversity being studied involves the absence of a structure or function (*Hochberg and Thornton, 2017*). Epistasis between loci presents another difficulty: when a putatively causal factor, such as a regulatory protein or DNA element, is swapped horizontally from one species into a model organism to test its causal role and recapitulates the phenotype from the 'source' species, this provides strong evidence for causality, but a negative result could arise because of an epistatic 'mismatch' between the swapped factor and other loci. The more distantly diverged the two species, the more significant this problem becomes (*Gehrke and Shubin, 2016*).

Ancestral sequence reconstruction (ASR) can, in principle, help to address these difficulties. ASR uses statistical phylogenetic methods to infer the sequences of ancient genes or proteins, which can then be synthesized and experimentally characterized (*Hochberg and Thornton, 2017*; *Harms and Thornton, 2010*). This approach allows the direction of evolutionary change to be determined and localized to a specific interval of phylogenetic history. The relatively small number of candidate mutations that occurred during this interval can then be introduced into the appropriate ancestral protein to directly test the hypothesis that they caused the evolutionary shift in biochemical properties and/or biological function. By using the ancestral background, this approach also minimizes the confounding effect of intragenic epistasis. ASR can also help to reduce the complications of intragenic epistasis that may affect horizontal transgenesis: the mismatch between the reconstructed ancestral protein and the genetic background of the model organism in which it is tested is less severe, because divergence between the source and the model occurred along only one branch from the ancestor, instead of two. Moreover, testing ancestral factors from successive phylogenetic nodes, which may differ by only a few sequence sites, drastically limits the scope of the inference that might be compromised by epistasis.

ASR is now widely used in evolutionary molecular biology and biochemistry (*Bridgham et al., 2009*; *Natarajan et al., 2013*; *Ortlund et al., 2007*; *Anderson et al., 2016*; *Bridgham et al., 2006*; *Finnigan et al., 2012*; *Hart et al., 2014*; *Howard et al., 2014*; *McKeown et al., 2014*;

*Nguyen et al., 2016*) and has been used to study the evolution of transcriptional networks in microbes (*Sorrells et al., 2015*; *Baker et al., 2012*; *Baker et al., 2013*). Recently, transgenic animals carrying reconstructed ancestral genes have been engineered to test the effects of historical genetic changes on physiology and fitness (*Siddiq et al., 2017a*), but this approach has not yet been applied to the study of development.

Here we use ASR to study the mechanisms for the historical evolution of a controlling molecular determinant of the anterior-posterior (AP) axis during embryonic development in flies (*Dearden and Akam, 1999*; *Patel, 1994*). In *Drosophila melanogaster* and other cyclorrhaphan flies – a large taxonomic group that appeared >140 Mya (*Grimaldi and Engel, 2005*) – AP patterning depends on an anterior gradient of Bicoid (Bcd), a transcription factor translated in the embryo from maternally deposited, anteriorly localized mRNA (*Driever and Nüsslein-Volhard, 1988a*; *Driever and Nüsslein-Volhard, 1988b*; *Berleth et al., 1988*; *Little and Wieschaus, 2011*). Embryos lacking Bcd fail to produce any head or thoracic structures, and duplicate posterior structures are formed in the embryo's anterior ([*Frohnhöfer and Nüsslein-Volhard, 1986*]; see also *Figure 1—figure supplement 1* panels K, L). Bcd directly regulates transcription of dozens of target genes (*McGregor et al., 2007*; *Chen et al., 2012*; *Driever et al., 1989*; *Struhl et al., 1989*) by using its homeodomain (HD) to bind specific cis-regulatory DNA sequences, particularly variants of the canonical motif TAATCC, with flanking sequences providing some additional specificity (*Driever et al., 1989*; *Struhl et al., 1989*; *Noyes et al., 2008*; *Treisman et al., 1989*; *Dave et al., 2000*). Bcd also binds the 3' untranslated region of the messenger RNA of Caudal (Cad), a transcription factor involved in development of the posterior end, and represses its translation (*Chan and Struhl, 1997*; *Rivera-Pomar et al., 1996*). In some insect taxa outside the Cyclorrhapha – such as beetles, wasps, and midges – the gene regulatory network that carries out AP development is similar to that in *D. melanogaster*, but its anterior deployment is controlled not by Bcd but by other upstream transcription factors (*Brent et al., 2007*; *Klomp et al., 2015*; *Lynch et al., 2006*).

Bcd is unique to cyclorrhaphan flies. In winged insects, the Hox3 gene is called Zen; a duplication of this gene deep in the cyclorrhaphan lineage yielded two paralogs, Bcd and Zen (*Stauber et al., 1999*; *Stauber et al., 2002*). Bcd's developmental role, anterior localization, and DNA specificity all appear to be derived evolutionary features, whereas many of Zen's appear to be ancestral (*Schmidt-Ott et al., 2010*). Specifically, neither Hox3 nor Zen is anteriorly localized in the embryo or involved in establishing AP polarity, and both seem to play conserved roles in extraembryonic tissue formation (*Rafiqi et al., 2008*; *Rafiqi et al., 2010*; *van der Zee et al., 2005*). Both Hox3 and Zen strongly prefer the consensus DNA motif TAATTA over Bcd's TAATCC consensus motif (*Noyes et al., 2008*; *Berger et al., 2008*), and neither regulates *Cad* translation (*Schoppmeier et al., 2009*). Evolution of Bcd's developmental role in cyclorrhaphan flies therefore required, at minimum, two major changes after its emergence by gene duplication: (1) Bcd must have evolved the capacity to regulate expression of TFs that specify anterior and posterior structures and (2) maternal localization of Bcd in the anterior of the egg must also have evolved. Changes in Bcd's target loci and in proteins with which Bcd interacts may also have contributed to the evolution of Bcd function. Here, we focus on the mechanisms that drove the first category of evolutionary changes: the acquisition by the Bcd protein of the capacity to regulate its target genes.

Numerous studies have addressed the sequence-structure function relations of Bcd and other HD proteins in *D. melanogaster*, but little is known about the mechanisms that drove the evolution of Bcd and its developmental role. Biochemical analyses in vitro suggest the hypothesis that acquisition of a lysine residue at position 50 (K50), which is located on the recognition helix (RH) that directly contacts the DNA major groove, may have played a role. K50 is present in several HD proteins that bind canonical target sites similar to Bcd's, but HDs that prefer Zen-like motifs contain a glutamine at this position (Q50). Biochemical studies have shown that swapping these two states between extant HDs affects relative affinity for these target sequences and suggest potential structural mechanisms for this effect (*Noyes et al., 2008*; *Treisman et al., 1989*; *Ades and Sauer, 1994*; *Hanes and Brent, 1989*; *Niessing et al., 2000*; *Percival-Smith et al., 1990*; *Schier and Gehring, 1993*; *Baird-Titus et al., 2006*; *Otting et al., 1990*). Other residues in the protein's recognition helix, including an arginine at site 54 (R54) have also been shown to affect DNA and RNA recognition in vitro (*Dave et al., 2000*; *Niessing et al., 2000*; *Hanes and Brent, 1991*). There has been no direct work, however, to assess what portion of the shift in DNA specificity that occurred during Bcd evolution is directly attributable to these or any other historical sequence substitutions.

In vivo, there is no evidence to indicate the extent to which Q50K or any other substitution can account for the evolution of Bcd's regulatory or developmental functions in the embryo. The only prior manipulative study to examine the in vivo effects of Q50K on Bcd function introduced this mutation into *D. melanogaster* Fushi tarazu (Ftz), a distantly related HD protein, and found that Q50K does not confer activation of any Bcd targets in fly embryos (*Schier and Gehring, 1993*) or cultured cells (*Zhao et al., 2000*). This observation could indicate that Q50K made no contribution to the evolution of Bcd's developmental functions; alternatively, it may have contributed to Bcd evolution, but only in combination with other Bcd-specific residues not introduced in this experiment; yet another possibility is that other characteristics of the present-day Ftz protein, not present in either Bcd or the Bcd-Zen ancestor, are incompatible with Bcd-like functions in the *D. melanogaster* embryo and therefore masked the contribution of Q50K in this experiment. A recent study found that replacing the HD of *D. melanogaster* Bcd with that of orthodenticle (*otd*), which also contains K50 and many other sequence differences, abolishes activation of most of Bcd's targets and does not rescue anterior development in transgenic embryos, suggesting that this residue is not sufficient to account for most of the regulatory functions of the Bcd HD (*Datta et al., 2018*). No studies have addressed the in vivo developmental effect of historically relevant substitutions at other sites on Bcd's functions.

To directly and explicitly test hypotheses about the effects of historical sequence changes on evolution of Bcd and its HD, we reconstructed ancestral forms of Bcd and used both in vitro and in vivo experiments to dissect the genetic and evolutionary trajectory of Bcd's functional evolution. This strategy allowed us to localize the change in Bcd function to a specific phylogenetic branch, quantify the biochemical and developmental aspects of that change, and characterize the extent to which historical amino acid substitutions within the HD, including but not limited to Q50K, were necessary and sufficient to account for the evolution of Bcd-specific functions.

## Results

### Developmental roles of Zen and Bcd homeodomains

Only the HD region of Bcd, Zen, and Hox3 retains sufficient phylogenetic signal to enable high-confidence ancestral sequence reconstruction. We therefore sought first to determine the extent to which the developmental functions of Bcd and Zen in extant *D.melanogaster* are attributable to differences between their HDs.

We established an in vivo gene replacement assay using *bcd*-null (*bcd^E1^*) *D. melanogaster* females and characterized both morphological and gene-regulatory phenotypes (*Figure 1*; *Figure 1—figure supplement 1*). As expected, *bcd*-null females produce embryos that lack all cephalic and thoracic structures; they have only one to three of the anterior-most abdominal segments and develop ectopic anterior filzkörper – posterior structures that are normally restricted to the eighth abdominal segment (*Frohnhöfer and Nüsslein-Volhard, 1986*) (*Figure 1—figure supplement 1K,L*). These embryos fail to activate transcription of known Bcd target genes, including *hb, kni, gt, otd, ems, btd*, and the first two stripes of *eve*; translational repression of *cad* in the anterior is also lost (*Figure 1—figure supplement 1M–T*). We introduced into *bcd*-null embryos constructs coding for full-length or chimeric versions of *D. melanogaster* Bcd or Zen under the control of the *bcd* promoter (BP) and the *bcd* 3'UTR (*Figure 1*; See Materials and methods). RNAs from all transgenes were anteriorly localized in a fashion indistinguishable from the endogenous *bcd* mRNA pattern (*Figure 1—figure supplement 1EE-GG*). We assessed morphological rescue by the following criteria: repression of ectopic anterior filzkörper, development of all eight abdominal segments, development of all three thoracic segments, and development of internal skeletal elements in the cephalic region. Target gene regulation was considered partially rescued if expression in the anterior could be detected by in situ hybridization, even if the patterns were reduced or spatially shifted.

As expected, the wild-type *D. melanogaster* Bcd transgene (DmBcd) rescued the *bcd^E1^* phenotype by all molecular and morphological criteria, and over 95% of embryos survived to adulthood (*Figure 1A–J*). In contrast, *Drosophila* Zen (DmZen) failed to rescue any aspect of the *bcd*-null phenotype (*Figure 1K–T* compared to *Figure 1—figure supplement 1K–T*). The Zen protein therefore shares few if any regulatory activities with Bcd, even when it is expressed in Bcd-like fashion.

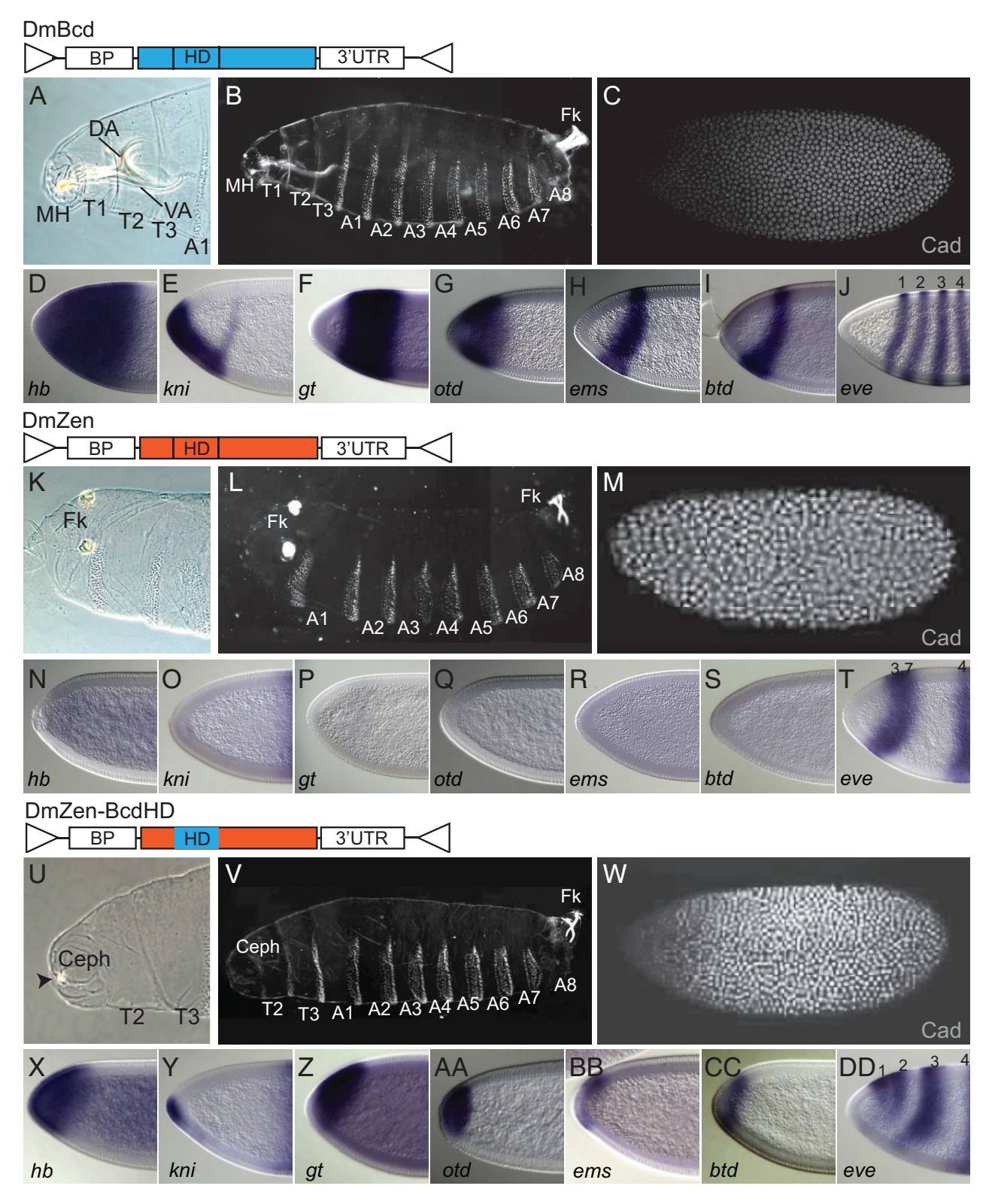

**Figure 1.** Role of the homeodomain in functional divergence of *D. melanogaster* Zen and Bcd. Offspring of transgenic *bcd*-deficient *D. melanogaster* containing two copies of various replacement transgenes are shown in cuticle preparations of larval head (A, K, U) and whole body (B, L, V); in situ hybridization experiments and antibody stains in transgenic embryos show expression of Bcd's translational (C, M, W) and transcriptional target genes (D–J, N–T, X–DD). Replacement transgenes were Bcd from *D. melanogaster* (DmBcd, A–J), Zen from *D. melanogaster* (DmZen, K–T), and a chimera of

*Figure 1 continued on next page*

*Figure 1 continued*

DmZen in which the homeodomain (HD) has been replaced by the HD of DmBcd (DmZen-BcdHD, **U–DD**). All three transgenes contain the *D. melanogaster bcd* promoter (BP), which directs maternal expression, and the *bcd* 3'UTR sequence required for anterior localization of the transgenic RNA. In the icons for each construct, protein-coding sequences from DmBcd are light blue and DmZen are red. Triangles show AttB sites used for recombination-mediated cassette exchange. For cuticle preps, mouthhooks (MH), dorsal arm (DA), ventral arm (VA), cephalic structures (Ceph), thoracic segments (T1–T3), abdominal segments (A1–A8), and Filzkörper (Fk) are indicated when present. An anti-Cad antibody was used to detect Cad protein. Antisense probes used in in situ hybridization experiments are indicated (*hb*: hunchback, *kni*: knirps, *gt*: giant, *otd*: orthodenticle, *ems*: empty-spiracle, *btd*: buttonhead, *eve*: even-skipped mRNAs). For comparison to wild-type, *bcd*-deficient, and a reciprocal construct of Bcd containing the ZenHD, see **Figure 1—figure supplement 1**.

DOI: https://doi.org/10.7554/eLife.34594.002

The following figure supplement is available for figure 1:

**Figure supplement 1.** Phenotypes of wildtype, *bcd*-null, and transgenic *D. melanogaster.*

DOI: https://doi.org/10.7554/eLife.34594.003

When DmZen was modified by replacing only its HD with that from *D.melanogaster* Bcd (DmZen-BcdHD, **Figure 1U–DD**), we observed substantial rescue of morphological and regulatory phenotypes. All thoracic and abdominal segments were formed, ectopic filzkörper structures were suppressed, and some head structures could be observed (**Figure 1U,V**). All seven of Bcd's transcriptional targets were activated, although their expression domains were shifted to the anterior (**Figure 1X–DD**), and Cad translation in the most anterior regions was partially repressed (**Figure 1W**). In contrast, a reciprocal construct of the Bcd coding sequence containing the Zen HD (DmBcd-ZenHD) conferred no rescue of any developmental or gene regulatory phenotypes (**Figure 1**, **Figure 1—figure supplement 1U–DD**). Sequence differences within the HD are therefore sufficient to explain a major portion of the Bcd protein's anterior patterning and gene regulatory activities in present-day *D. melanogaster* embryos, with much smaller contributions from differences elsewhere in the protein.

## Reconstruction of ancestral homeodomains

To understand when and how the Bcd HD's functions evolved, we reconstructed the phylogenetic history of the Bcd and Zen HDs and inferred ancestral sequences. The protein family phylogeny was inferred from an alignment of 33 Bcd, Zen, and Hox3 HD sequences from cyclorrhaphan flies and other insects (**Figure 2**; **Supplementary file 1**). Branch lengths and model parameters were estimated by maximum likelihood; topological relationships were inferred by maximum likelihood (**Figure 2—figure supplement 1**) and by constraining the topology to reflect relationships among species established in prior studies (**Figure 2**). As expected, both phylogenies indicate that duplication of Hox3/Zen before the radiation of the Cyclorrhapha produced two new paralogous genes – one lineage leading to cyclorrhaphan Zen and the other to Bcd. The phylogenies' branch lengths suggest that a burst of rapid evolution occurred in the Bcd HD after this duplication, whereas Zen's HD continued to evolve at a slower rate (**Figure 2**; **Figure 2—figure supplement 1**).

Using this alignment and the species-constrained phylogeny, we inferred the maximum likelihood (ML) amino acid sequence of the HD at two key ancestral nodes: (1) AncZB, the common ancestor of all Zen and Bcd proteins in cyclorrhaphan flies, which represents the single ancestral sequence that existed just before the duplication and (2) AncBcd, one of AncZB's daughter nodes, which represents the Bcd protein in the last common ancestor of all cyclorrhaphan flies (**Figure 2A**). Both sequences are reconstructed with little statistical ambiguity (mean posterior probability = 0.970 and 0.986, with only 3 and 2 sites ambiguously reconstructed in AncZB and AncBcd, respectively (**Supplementary file 2**). AncZB and AncBcd HD differ at 31 amino acid positions (**Figure 2B**).

## Divergence of DNA-binding preferences between AncBcd and AncZB

We first characterized the evolutionary changes in DNA specificity that occurred in the Bcd gene after its birth by duplication of AncZB. We synthesized coding sequences for AncZB and AncBcd HDs and characterized each protein's global DNA specificity using protein-binding microarrays (PBMs [*Berger et al., 2006*]) consisting of all possible DNA 10-mers (**Figure 3**). AncBcd's overall occupancy of DNA motifs was very similar to that of *D. melanogaster* Bcd (r = 0.94, **Figure 3A**). In contrast, the correlation between the sequences bound by AncBcd and those bound by AncZB was

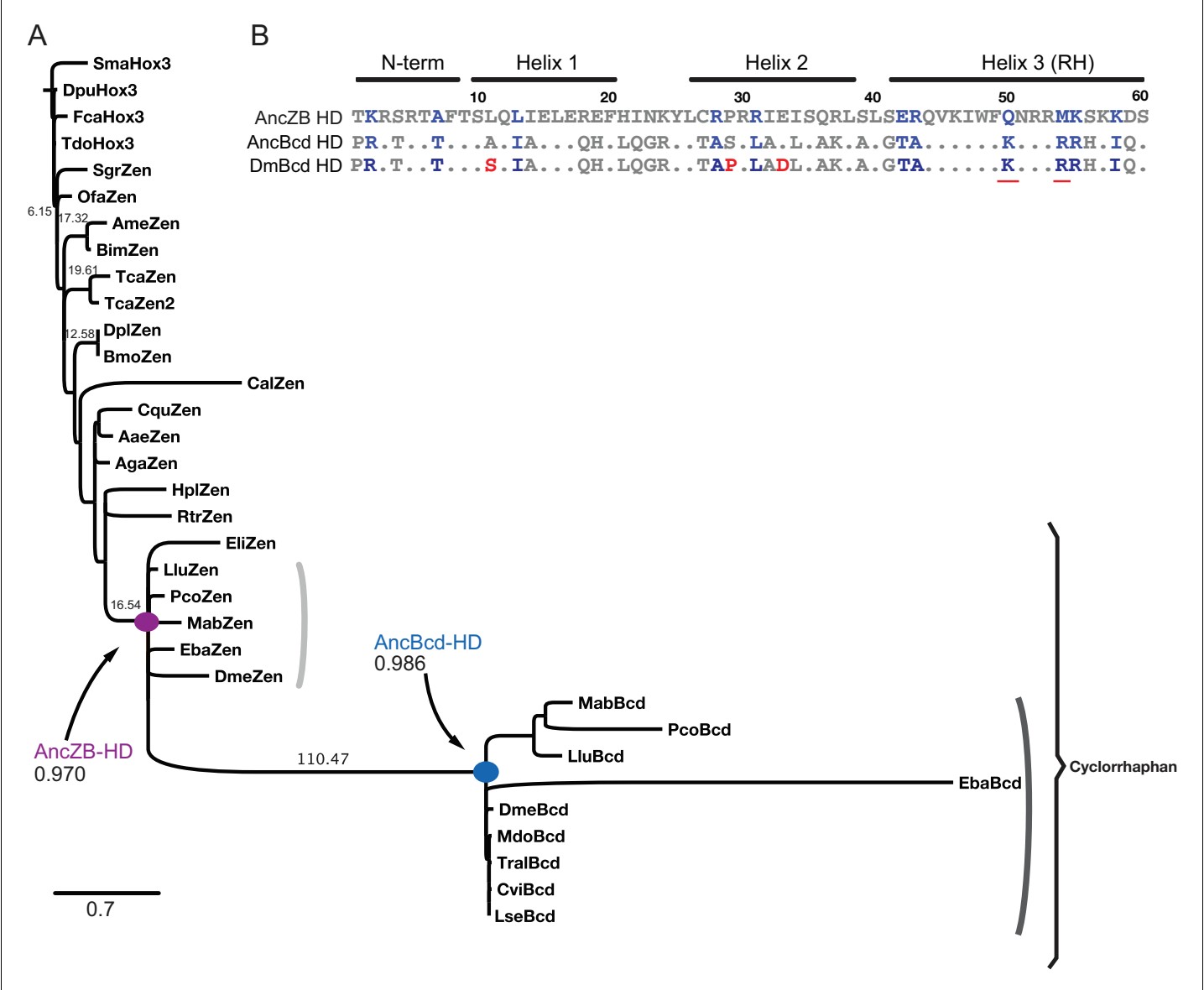

**Figure 2.** Reconstruction of ancestral HD sequences. (**A**) Phylogeny of Bcd, Zen, and Hox3 HD amino acid sequences from insects and other arthropods. One representative sequence is shown among sequence entries with no sequence diversity. Reconstructed ancestral nodes are labeled; the mean posterior probability across sites for each maximum likelihood ancestral sequence is shown. Branch support labels show the approximate likelihood ratio statistic. Scale bar, 0.7 amino acid changes per site. The tree was constrained to reproduce well-corroborated species relationships; for the maximum likelihood phylogeny, see *Figure 2—figure supplement 1*. For species names and sequences, see *Supplementary file 1*. (**B**) Amino acid sequences of AncZB, AncBcd and Bcd from *D. melanogaster* (Dm) HDs. For AncBcd and DmBcd, only residues that differ from the AncZB HD are shown. Of these, 11 substitutions (shown in blue) are diagnostic differences conserved among all sequences in the Bcd vs. Zen clades. Differences between AncBcd and DmBcd are labeled red. For site-specific posterior probabilities of ancestral states, see *Supplementary file 2*.
DOI: https://doi.org/10.7554/eLife.34594.004

The following figure supplement is available for figure 2:

**Figure supplement 1.** Unconstrained phylogeny.
DOI: https://doi.org/10.7554/eLife.34594.005

far weaker (r = 0.56, *Figure 3B*), with thousands of target sequences changing in relative occupancy between AncZB and AncBcd. Bcd's present-day global specificity is therefore a derived state that evolved on the phylogenetic branch after duplication of AncZB but before the last common ancestor of cyclorrhaphan flies.

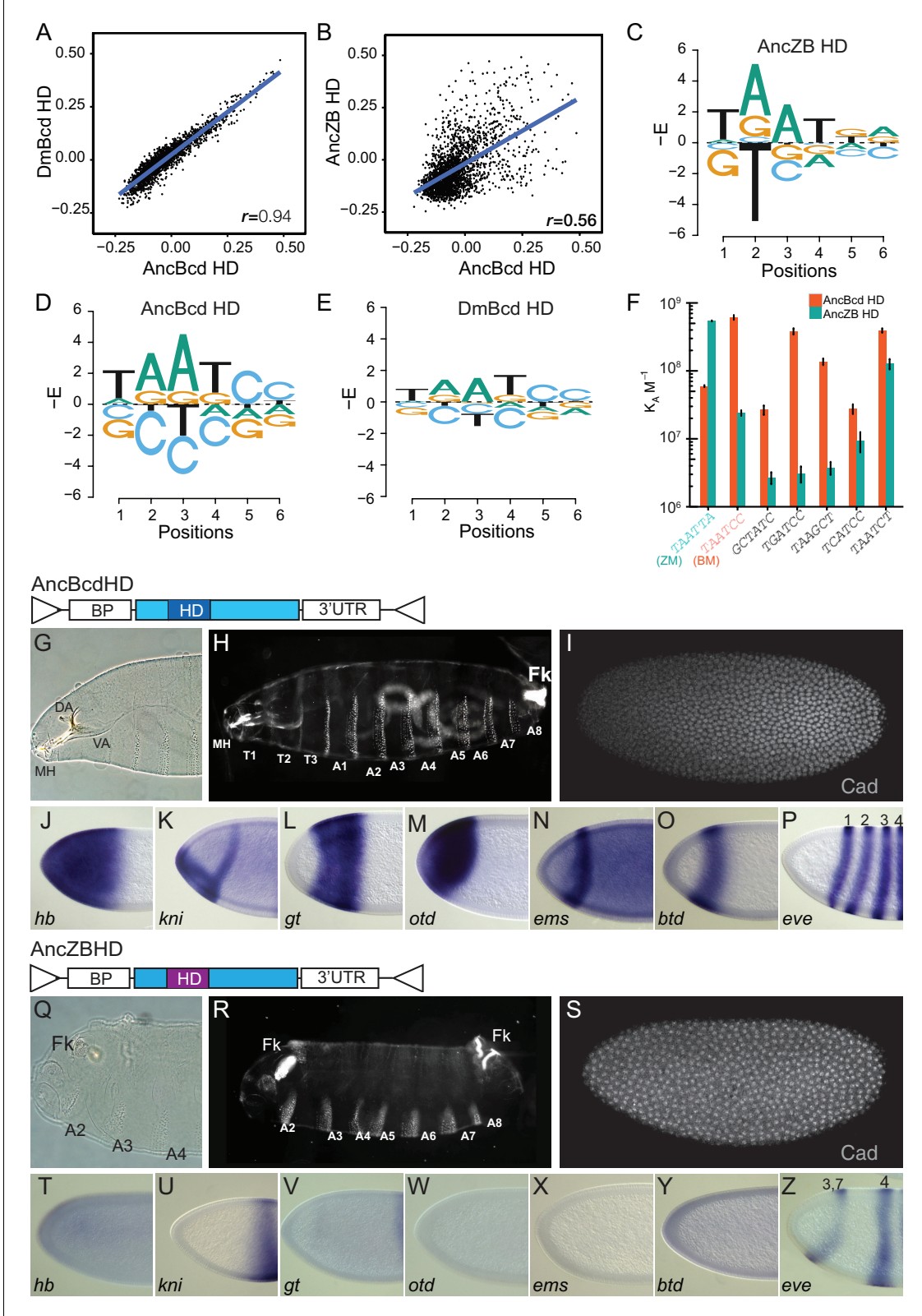

**Figure 3.** In vitro and in vivo functions of AncBcd and AncZB HDs. (A, B) Protein binding microarray (PBM) analysis of the DNA specificity of AncBcd, DmBcd, and AncZB HDs. Each dot represents one DNA 6-mer, plotted according to its binding enrichment (E-scores) in PBMs performed with two different HDs; Pearson's correlation coefficient for each comparison is shown. (C–E) Energy logos showing the site-specific sequence determinants of DNA binding by AncZB, AncBcd, and DmBcd HDs as determined from PBM data. The size of each letter is proportional to the absolute value of that

*Figure 3 continued on next page*

*Figure 3 continued*

base's main effect on the energy of binding relative to the mean of all bases at that position. The y-axis is unitless after division by *RT*. (F) Affinity of ancestral HDs for consensus Zen and Bcd DNA motifs (ZM and BM, sequences shown) in a quantitative fluorescence electrophoretic mobility shift assay. Several noncanonical Bcd targets previously shown to bind *Drosophila* Bcd were also tested. Column heights and error bars show the mean and SEM for three replicates. (G–Z) Cuticle phenotypes and gene expression patterns for offspring produced by *bcd*[E1] females carrying two copies of a rescue construct expressing *Dm* Bcd protein (light blue) in which the HD has been replaced by the reconstructed AncBcd HD (dark blue, panels G-P) or the reconstructed AncZB HD (purple, panels Q-Z). Individual panels and constructs are labeled as in *Figure 1*. For replicates and alternative ancestral reconstructions, see *Figure 3—figure supplement 1*, *Figure 4—figure supplement 1* and *Supplementary files 3* and *4*. For SPR analysis, see *Figure 3—figure supplement 2* and *Supplementary file 4*.

DOI: https://doi.org/10.7554/eLife.34594.006

The following figure supplements are available for figure 3:

**Figure supplement 1.** Binding site preference is robust to uncertainty about the ancestral sequence reconstruction.

DOI: https://doi.org/10.7554/eLife.34594.007

**Figure supplement 2.** Bcd evolved novel binding specificity by changes in sequence-specific dissociation rates.

DOI: https://doi.org/10.7554/eLife.34594.008

To characterize the evolutionary shift in preference for DNA motifs, we computed from the PBM data the additive effect of each possible nucleotide at each sequence position on the estimated binding energy of the protein-DNA complex (*Figure 3C,D,E*). AncZB, AncBcd, and DmBcd all share the core consensus motif TAAT (positions 1–4), but with some degeneracy. The major differences are at positions 5 and 6: AncZB prefers T or G at position 5 and A or G at position 6 (*Figure 3C*), but AncBcd and DmBcd strongly prefer sequences containing CC at these two positions (*Figure 3D, E*). An additional difference is that AncZB strongly excludes binding to motifs with T at position 2, whereas AncBcd excludes binding to C at this position (*Figure 3C,D*). These findings were robust to uncertainty about the sequences of the ancestral HDs: PBMs performed using alternative reconstructions of AncBcd and AncZB, which incorporated plausible alternative amino acid states at all ambiguously reconstructed sites (*Eick et al., 2017*), yielded almost identical results (*Supplementary file 3*).

To quantitatively characterize evolutionary changes in DNA preference that occurred between AncZB and AncBcd HDs, we used electrophoretic mobility shift assays (F-EMSA) to measure binding affinity to the PBM consensus Zen motif (ZM, TAATTA), the consensus Bcd motif (BM, TAATCC) and five other noncanonical motifs to which Bcd has previously been shown to bind (*Dave et al., 2000*; *Hanes and Brent, 1991*; *Zhao et al., 2000*; *Driever and Nüsslein-Volhard, 1989*; *Rivera-Pomar et al., 1995*; *Ma et al., 1996*). We found that relative preference for Bcd motifs increased dramatically across the interval from AncZB and AncBcd, with affinity for BM increasing 25-fold and that for ZM declining by a factor of 10 (*Figure 3F*). Binding to the noncanonical motifs also increased substantially from AncZB to AncBcd. Alternative reconstructions of AncZB and AncBcd HDs yielded nearly identical results (*Figure 3—figure supplement 1*).

We further investigated DNA binding by the ancestral proteins using a surface plasmon resonance assay (SPR) (*Figure 3—figure supplement 2*). Corroborating the results from EMSA, AncZB's affinity for ZM is higher than for BM, whereas AncBcd prefers BM. Changes in affinity were caused primarily by changes in dissociation rates: AncBcd's off-rate from BM was 66-fold slower than AncZB's, and its off-rate from ZM was three times faster; changes in on-rates were small (*Supplementary file 4*). As a result, there was a strong evolutionary shift in average residence times on the two motifs: between AncZB and AncBcd, residence time on BM increased by 70-fold, while that on ZM declined by a factor of 3 (*Figure 3—figure supplement 2*, *Supplementary file 4*). Although changes in preference were qualitatively similar when assessed by F-EMSA and SPR, the dissociation constants estimated by the two methods were somewhat different, presumably because buffers differ between the two techniques and because EMSA, unlike SPR, is a nonequilibrium assay.

Taken together, these data indicate that during the phylogenetic interval immediately after the duplication of AncZB, a major change in DNA recognition and affinity occurred along the lineage leading to AncBcd. The ensemble of occupied DNA binding sites shifted dramatically, and the protein's preference for canonical Zen vs. Bicoid motifs was inverted.

## In vivo characterization of ancestral HDs

We next tested the effects of the AncZB and AncBcd HDs on development and gene regulation in vivo. We swapped these domains precisely into the DmBcd rescue transgene and tested whether they rescue the defects of embryos from *bcd*-null females. The reconstructed AncBcd HD yielded a phenotype indistinguishable from that of the DmBcd transgene, with essentially complete morphological rescue (*Figure 3G,H*), activation of all Bcd targets (*Figure 3J–P*), and repression of Cad (*Figure 3I*), with 37% of embryos surviving to adulthood. In contrast, when the AncZB HD was swapped into the same construct, no morphological or molecular rescue was observed (*Figure 3Q–Z*). The derived in vivo functions of the Bcd homeodomain – including its capacity to regulate key developmental target genes and to specify formation of cephalic and thoracic structures in anterior regions of the embryo – therefore evolved almost entirely during the interval after duplication of AncZB but before AncBcd.

## Genetic causes for the evolution of AncBcd's DNA specificity

The genetic causes for the evolutionary change in the Bcd HD's functions must be among the 31 amino acid substitutions that occurred between AncZB and AncBcd (*Figure 2B*). Although previous research into the genetics of Bcd function focused on specific candidate residues, we took an unbiased approach. To identify the most likely causal changes, we examined phylogenetic patterns of conservation and divergence. Eleven of the 31 amino acid substitutions that occurred between AncZB and AncBcd are phylogenetically diagnostic, with the AncBcd state conserved in all descendant Bcd proteins (*Figure 2B*); these diagnostic substitutions constitute an initial set of candidate causal changes.

We first tested whether each of these 11 substitutions was necessary for the evolution of AncBcd's preference for the canonical Bicoid motif (BM). We individually reverted each diagnostic residue in AncBcd HD to the state present in AncZB HD and measured the effect on DNA affinity using F-EMSA. Only one reversal – changing lysine 50 back to the ancestral glutamine (K50q) – shifted the protein's preference for BM back toward ZM; this reversion caused a 38-fold change in relative affinity and restored the ancestral preference for ZM. Mutating arginine (R) to the ancestral methionine (m) at position 54 (R54m) – had no effect on relative preference, but it lowered affinity to both BM and ZM by a factor of ~15 (*Figure 4A*). No other ancestral state affected affinity to either BM or ZM or relative preference by a factor of three or more in either direction, indicating that none of these were necessary for the historical evolution of the derived DNA preference of the Bcd HD.

To test whether the substitutions at positions 50 and 54 were sufficient to cause the evolutionary change in DNA recognition in vitro, we introduced the derived states from AncBcd into the AncZB HD (*Figure 4B*). Substitution q50K had a very large effect, shifting the ancestral protein's relative binding preference by four orders of magnitude, increasing affinity for BM 100-fold and reducing that for ZM by a factor of 100 (*Figure 4B*). By comparison, simultaneously introducing into AncZB all 30 other substitutions from this interval except q50K (construct AncBcd-K50q) caused no improvement in affinity for BM and just a seven-fold reduction in affinity to ZM (*Figure 4A*). Substitution m54R alone increased affinity for both ZM and BM, but in the presence of q50K, it had no significant effect, indicating an epistatic interaction by which q50K silences the effect of m54R (*Figure 4B*). These data indicate that the single substitution q50K was sufficient to cause the entire historical shift in in vitro DNA preference from ZM to BM, whereas m54R did not make an observable contribution to the new specificity.

We next used PBMs to determine whether the evolution of Bcd's global DNA specificity was also attributable primarily to substitution q50K. We compared the ensemble of DNA motifs bound by AncZB to those bound by AncZB-q50K. This substitution shifted the overall enrichment and de-enrichment across all oligomers to be extremely similar to those bound by AncBcd (*Figure 4C*, r = 0.93, compared to r = 0.56 for the correlation of AncZB and AncBcd (*Figure 3B*). Substitution q50K also transformed the binding-energy logo of AncZB into one almost identical to that of AncBcd, with the 'CC' dinucleotide strongly favored at positions 5 and 6 and the ancestrally preferred bases now de-enriched (*Figure 4D*, compared to *Figure 3D*). Substitution q50K was therefore sufficient to confer on AncZB the derived in vitro DNA-binding profile of AncBcd.

The other HD substitutions did not contribute to the evolutionary shift in global DNA specificity, as shown by a PBM comparison of AncZB to AncBcd-K50q, which is AncZB plus all 30 other derived

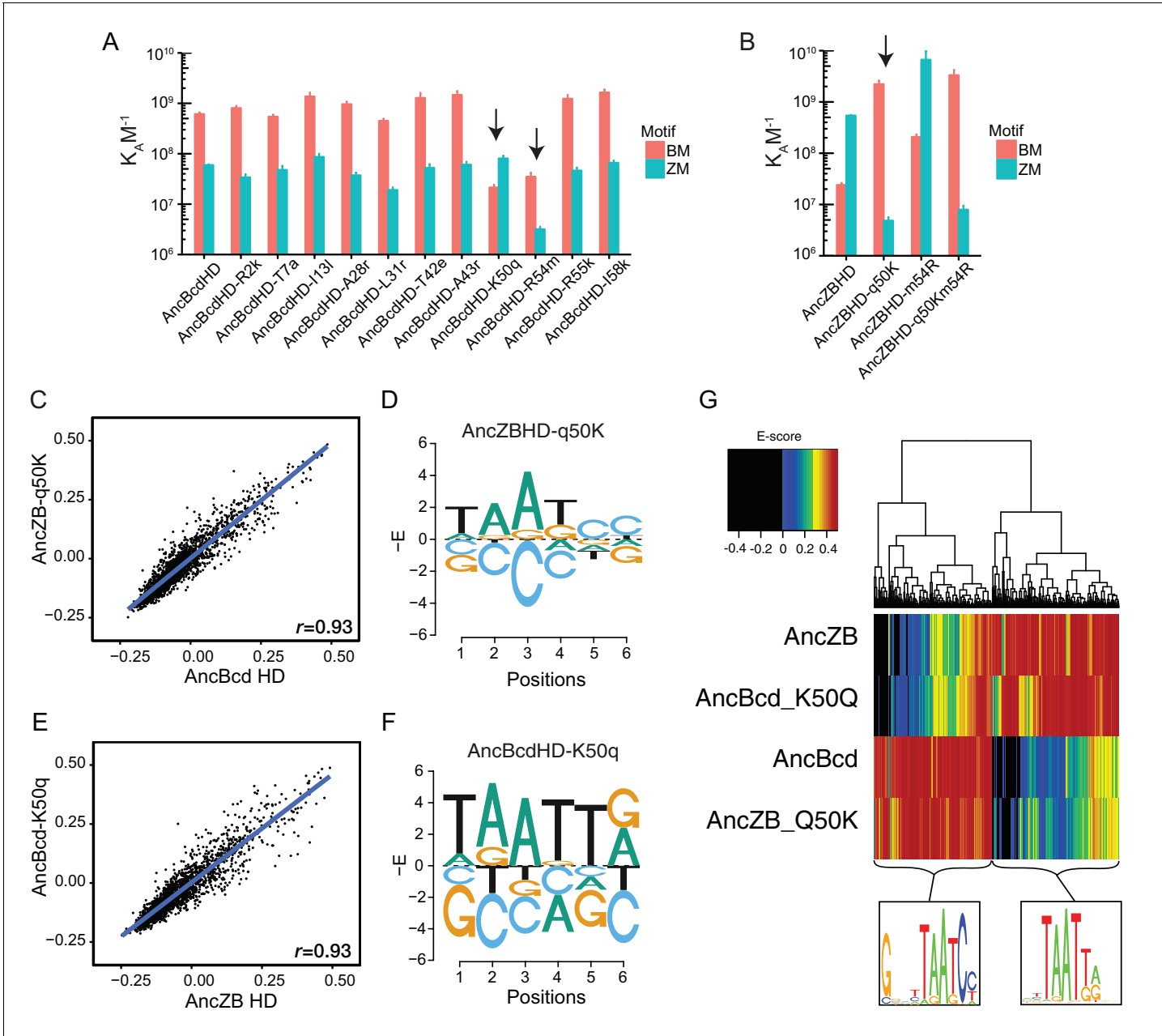

**Figure 4.** A single amino acid substitution changes AncZB's in vitro binding preference. (**A**) Effect of reversing diagnostic historical substitutions on AncBcd-HD's affinity for consensus Bcd and Zen motifs (BM and ZM), in a fluorescent EMSA assay. Column height and error bars show mean and SEM of three replicates. Upper and lower cases denote derived and ancestral states, respectively. (**B**) Effect of introducing historical substitutions to the derived state at site 50 and 54 into AncZB HD. (**C**) Binding of DNA motifs by AncZBHD-q50K is tightly correlated with binding by AncBcdHD. Each point is a DNA 6-mer plotted by the average of E-scores for 8-mers containing that 6-mer. Pearson correlation coefficient is shown. (**E**) Binding of DNA motifs by AncBcdHD-K50q is tightly correlated with binding by AncZB-HD. (**D, F**) The PBM energy logos of AncZBHD-q50K and AncBcdHD-K50q (compare to *Figure 3C*). (**G**) Effect of historical substitutions on global binding specificities in PBMs. Each column represents an 8-mer, colored by its PBM E-score, according to the color scale shown. Each row shows PBM of a reconstructed HD protein with or without residue 50 swapped between ancestral and derived states. 8-mers were clustered using the Manhattan distance metric. Only 8-mers with E-score ≥ 0.45 for at least one HD variant are shown. Logos at the bottom of the heatmap were created from the 8-mers in the two large clusters.

DOI: https://doi.org/10.7554/eLife.34594.009

The following figure supplement is available for figure 4:

**Figure supplement 1.** 8mer binding profiles for replicate HD constructs.

DOI: https://doi.org/10.7554/eLife.34594.010

states that evolved in the AncZB-AncBcd interval (except for q50K). The global DNA specificity of this protein remained strongly correlated to that of AncZB (r = 0.93, *Figure 4E*); its binding-energy motif was also like that of AncZB, with a strong preference for G/T and A at the last two positions, and depletion of binding to the CC bases that are enriched among AncBcd's targets (*Figure 4F*). The overall profiles of DNA targets bound by AncZB and AncBcd-K50q are highly similar, with enrichment of sequences similar to TAATTA and de-enrichment of those similar to TAATCC; conversely, AncBcd and AncBcdq50K show the opposite pattern (*Figure 4G*).

Taken together, these results indicate that substitution q50K was the primary determinant of the evolutionary shift in in vitro DNA-binding preferences that occurred during the transition from AncZB to AncBcd. The other 30 substitutions that occurred on this branch, including m54R, had very minor effects on the in vitro DNA-binding specificity of the evolving HD.

## In vivo effects of historical substitutions on gene expression and development

To test the contribution of historical substitutions to the evolution of Bcd's role in AP specification and the regulatory network that drives it, we incorporated candidate historical residues into transgenic Bcd constructs containing the AncZB or AncBcd HDs, transformed *bcd-null* females with these constructs, and assayed their offspring.

We found that reversing only q50K (AncBcdHD-K50q) completely abolished the rescue conferred by AncBcd HD: larval cuticles were similar to *bcd-null* embryos (*Figure 5A,B*), expression of Bcd's transcriptional targets was completely lost (*Figure 5D–J*), and Cad translation was no longer repressed (*Figure 5C*). Substitution q50K was therefore necessary for evolution of the developmental and gene regulatory functions of the Bcd HD.

To determine whether q50K was also a sufficient historical cause, we introduced the derived lysine into the AncZB HD. We observed partial rescue (*Figure 5K–T*). Ectopic filzkörper formation was suppressed, and all eight abdominal segments were formed in ~90% of the embryos (*Supplementary file 5*). The posterior-most thoracic segment (T3) was observed in around 60% of larvae. These embryos failed to form anterior thoracic segments (T1 and T2) and any head structures (*Figure 5K,L*, *Supplementary file 5*). Of Bcd's seven transcriptional targets, three (*hb, kni,* and *eve stripe 2*) were activated, but four (*gt, otd, ems,* and *btd*) were not (*Figure 5N–T*), and Cad translation was not repressed (*Figure 5M*). Thus, q50K partially accounts for the evolution of Bcd's role in development and gene regulation in vivo.

We therefore examined whether substitution m54R – the only other historical substitution with a significant effect on DNA binding in vitro – affected the functions of the ancestral proteins. Introducing m54R alone into AncZB HD conferred no observable rescue of developmental or gene regulatory phenotypes (*Figure 5U–DD*). Introducing m54R together with q50K, however, yielded a notably more complete rescue than q50K alone: all larvae carrying the double substitution contained T3 and all eight abdominal segments,>80% also contained T2, and >50% formed anterior structures similar to the sclerotized structures of the internal cephalic skeleton (*Figure 5EE,FF*; *Supplementary file 5*). Compared to the single q50K substitution, two additional Bcd-dependent transcriptional targets, *gt* and *otd*, were activated (*Figure 5HH–KK,NN*), but two others, *btd* and *ems*, remained silent (*Figure 5LL,MM*), and Cad translation was still not repressed in the anterior (*Figure 5GG*). These data point to an epistatic interaction between m54R and q50K in vivo, in which m54R has an effect only when introduced together with q50K; this interaction differs from that observed in vitro, where m54R affects affinity in isolation but not in combination with q50K.

Thus, just two large-effect substitutions – q50K and m54R – are sufficient in the ancestral HD to recapitulate the evolution of major aspects of the Bcd HD's derived function, including regulation of many target genes and partial specification of anterior structures. Compared to the phenotype of AncBcd-HD, the rescue conferred by these two substitutions is incomplete, with several targets remaining unregulated and anterior development still compromised. Additional contributions must therefore have been made by some of the other historical substitutions in the HD during the interval between AncZB and AncBcd.

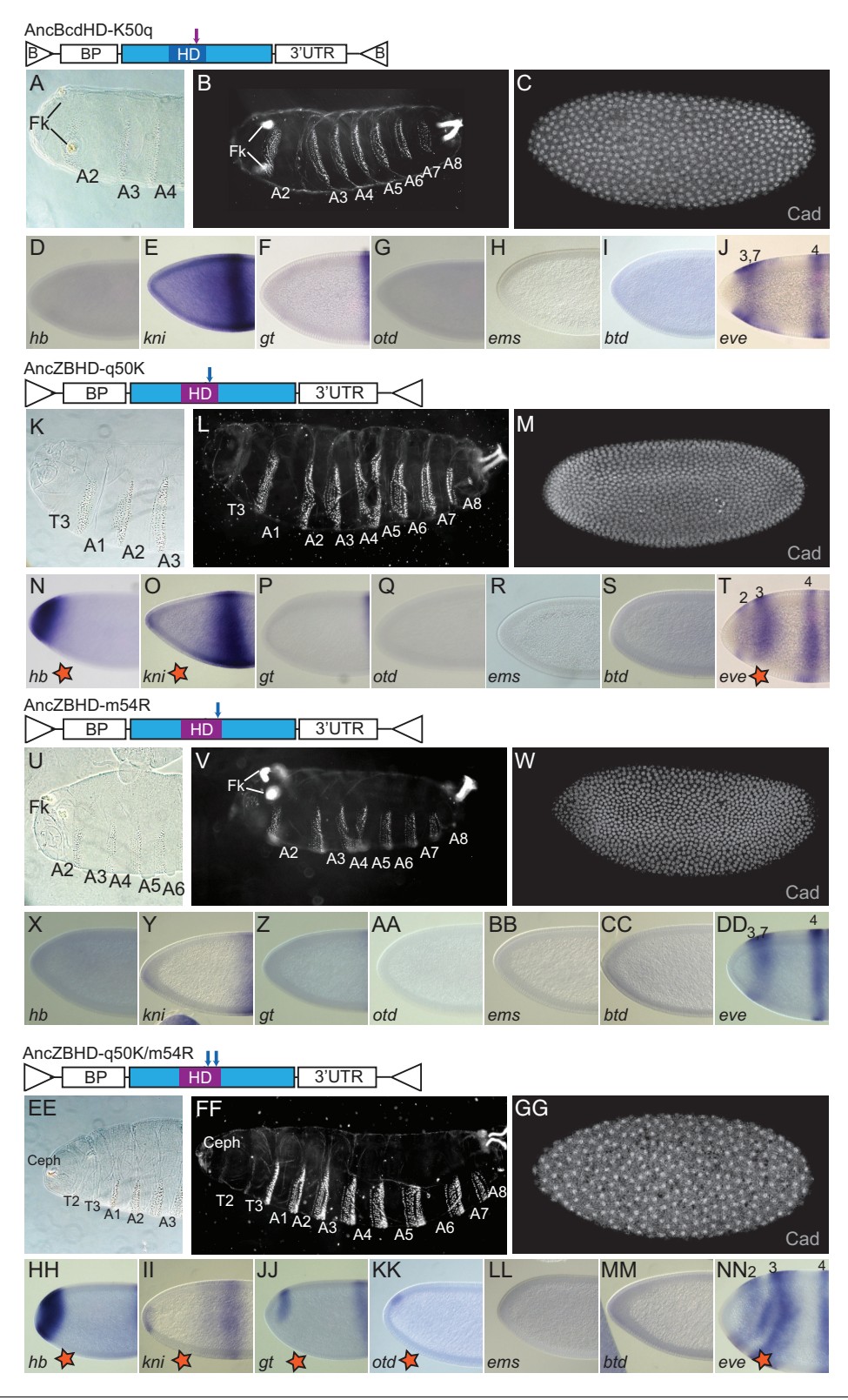

**Figure 5.** Effect of historical substitutions on Bcd functions in vivo. (**A–J**) Testing the necessity of q50K for AncBcd function. Cuticle phenotypes and Bcd-target gene expression patterns of embryos from *bcd*[E1] females carrying two copies of a Bcd rescue construct containing AncBcd-HD (dark blue) with an amino acid reversion to the AncZB ancestral state at residue 50 (purple arrow). (**K–NN**) Testing the sufficiency of historical substitutions of q50K and m54R. Embryos expressing Bcd rescue constructs containing AncZB-HD (purple) with historical substitutions to the derived AncBcd state (blue

*Figure 5 continued on next page*

*Figure 5 continued*
arrows) at site 50 (**K–T**), 54 (**U–DD**), or both (**EE–NN**). Individual panels and constructs are labeled as in *Figure 1*. Stars indicate Bcd-dependent target gene activation induced by the indicated constructs.
DOI: https://doi.org/10.7554/eLife.34594.011

## Discussion

### Relation to prior work

The evolution of a molecule and its functions can be conceived of as a specific historical trajectory through the vast space of possible sequences from the ancestral to the derived form. A complete understanding of the evolution of a protein's role in development would identify the sequence substitutions that changed its functions, order them in a temporal sequence, and characterize how each change altered specific molecular and developmental processes. This 'vertical' approach is different from that used by most studies, which compare the molecular sequences and developmental functions of present-day proteins. For example, experiments that perturb the sequence of a present-day protein, introducing amino acid states that have different biochemical properties or are found in other proteins that have different functions, explore the local sequence space immediately surrounding that protein. Although this approach can help to identify some of the genetic and biochemical constraints that shape a molecule's current functions, it does not directly address the causal effects of the set of historical changes in gene sequence that occurred during its evolution. (*Hochberg and Thornton, 2017*)

Previous work investigated the genetic determinants of *D. melanogaster* Bcd's functions using non-vertical approaches. In vitro experiments have shown that mutating Q50 to K in distantly related extant HDs increases affinity for or activation from consensus Bcd motifs and reduces it from Zen motifs; the converse experiment in DmBcd HD has the opposite effect (*Treisman et al., 1989*; *Ades and Sauer, 1994*; *Percival-Smith et al., 1990*; *Hanes and Brent, 1991*; *Hanes and Brent, 1991*; *Zhao et al., 2000*). Mutations at some other sites, including position 54, also affect affinity and/or preference (*Dave et al., 2000*; *Niessing et al., 2000*; *Damante et al., 1996*). These studies established that certain residues are important for the present-day protein's functions, but their design could not characterize how much of the evolutionary acquisition of Bcd's functions was caused by these or any other historical sequence substitutions. First, they provided no evolutionary scale to refer to: the quantitative change in function that occurred during the key evolutionary interval was unknown, so how much of that change was attributable to Q50K or other substitutions could not be assessed. Second, most mutation studies (with the exception of Q50K) exchanged amino acids for non-ancestral states that never occurred during Bcd evolution. For example, mutating Bcd's Arg54 to alanine, serine, or lysine affects in vitro functions, as does introducing Arg54 into distantly related HDs that contain Ala or Tyr (*Dave et al., 2000*; *Niessing et al., 2000*; *Damante et al., 1996*; *Pellizzari et al., 1997*); however, none of these amino acids were present in the ancestral HD, so these results provided no information about the impacts of the Met-Arg substitution that occurred at this site during Bcd evolution. Indeed, without a phylogenetic analysis, the substitutions that occurred during history– the set of transitions from ancestral to derived amino acids that temporally co-occurred with the change in Bcd function – had not been identified. Third, even when the relevant derived and ancestral states were studied, their effects were not assayed in the appropriate historical sequence context, a critical issue because mutations can have different effects on function in extant vs. ancestral backgrounds (*Hochberg and Thornton, 2017*). Thus, it has been unknown whether any historical substitution – even the well-studied position 50 – was an entirely sufficient cause, a partial contributor, an epistatic modifier, or causally irrelevant in the historical evolution of Bcd's derived functions in vitro.

Even less was previously known about the genetic causes of the evolution of Bcd's developmental and gene regulatory functions in vivo. Mutating Q50 to K in the HD of the distantly related protein Ftz failed to confer activation of Bcd-responsive genes in transgenic fly embryos or in cultured cells (*Schier and Gehring, 1993*; *Zhao et al., 2000*). This negative result might imply that Q50K did not cause the evolution of any of Bcd's derived functions in vivo, but it could also be an artifact of epistasis, if other sequence states in Ftz are incompatible with Bcd functions but were not present in the

ancestral Bcd protein when the substitution occurred. Q50K has also been introduced into other HDs, but effects on other developmental processes, not on *Drosophila* anterior development or expression of Bcd-responsive genes, were studied (*Ibrahim et al., 2013*; *Percival-Smith et al., 1997*; *Capovilla et al., 1994*; *Plaza et al., 2001*) At site 54, the mutations studied in vivo have involved states that did not occur during Bcd evolution, such as alanine and serine (*Niessing et al., 2000*), so the effects of M54R on its own, or in combination with Q50K, have been unknown.

Our strategy was designed to directly characterize the causal contribution of historical substitutions to the acquisition of Bcd's derived functions, both in vitro and in vivo, using the reconstructed context of the ancestral proteins in which they occurred. By functionally characterizing successive ancestral nodes on the phylogeny, we found that all the derived functions of the *Drosophila* Bcd HD were acquired on one phylogenetic branch, which represents the evolutionary interval between the duplication of AncZB, which produced a separate *bcd* locus, and before the *bcd* gene in the last common ancestor of all cyclorrhaphan flies (AncBcd). During this interval, a dramatic shift occurred in the global set of DNA sequences to which AncBcd binds, and the relative preference for canonical BMs vs. ZMs changed by a factor of 250. This interval was also the key period for the evolution of Bcd's in vivo functions: whereas AncZB-HD rescued no Bcd-specific functions, AncBcd-HD activated all seven Bcd target genes tested, inhibited *Cad* translation, and organized development of many anterior structures in the embryo, yielding functions and phenotypes indistinguishable from those conferred by the *D. melanogaster* Bcd HD.

The sequence changes that occurred along this phylogenetic branch represent the candidate genetic causes for the evolution of Bcd HD's derived functions. We focused on the contribution of 11 substitutions from this interval, because the derived states of these are conserved among all known Bcd proteins. When each of these was introduced singly into AncZB, only Q50K made a measurable contribution to the new specificity, and this substitution alone was sufficient to recapitulate the entire evolutionary shift in local specificity (affinity for BM and ZM) and global DNA affinity (as measured in a PBM), yielding in vitro functions indistinguishable from AncBcd. The other substitutions had no effect on motif preference, and only M54R had a significant effect on affinity, conferring a nonspecific increase in binding to both motifs; however, when M54R was introduced together with Q50K, it had no measurable effect on affinity for either motif, compared to Q50K alone. Thus, a single large-effect substitution appears to provide a nearly complete causal explanation for the evolution of the Bcd HD's derived DNA specificity as measured in our in vitro assays.

By performing transgenic experiments with reconstructed ancestral HDs, we sought to directly assess the contribution of historical substitutions to the shift in embryonic gene expression and developmental phenotypes that occurred during Bcd evolution. We found that Q50K was a major evolutionary cause: introducing it into AncZB-HD conferred regulation of three of seven Bcd target genes and recovery of some aspects of anterior development. Unlike its minor role in vitro, M54R was an important causal contributor in vivo, conferring regulation of two additional targets and a more complete developmental rescue, but only when introduced together with Q50K. At least some of the 29 other substitutions that occurred during this interval made additional contributions, conferring activation of the last two transcriptional targets (*ems* and *btd*), increasing expression levels and sharpening localization of the entire suite of targets, yielding anterior repression of *Cad* translation, and rescuing the full anterior developmental phenotype of AncBcd-HD. Further work will be required to determine the specific contribution of these substitutions to the evolution of Bcd function.

## Limitations

There are several limitations to our approach, but none are likely to undermine our major claims. First, ancestral sequences are inferred statistically and are therefore never known with certainty. The ML sequences of AncZB and AncBcd were inferred with very high confidence, with only 3 and 2 sites, respectively, having a plausible alternative amino acid, and the expected number of erroneously reconstructed sites is just 1.7 and 0.9 in the two proteins. We performed the in vitro assays using 'AltAll' versions of the reconstructed proteins that contain all plausible alternative residues and found no differences in function from the ML ancestors. Our inferences concerning ancestral functions therefore appear to be robust to stochastic uncertainty about the precise sequence of the ancestral proteins. We did not test these alternative reconstructions in transgenic animals and therefore cannot strictly rule out the possibility that the true ancestral sequences of AncZB and AncBcd

may have functioned differently from the ML versions in vivo. But this possibility seems unlikely, given the high degree of statistical confidence and the robustness of the protein's in vitro functions to sequence uncertainty.

A second limitation is that we assayed reconstructed ancestral HDs in the context of an extant model organism. The developmental gene regulatory environment is likely to have changed in some ways from the ancestral cyclorrhaphan to *D. melanogaster*; it is possible that the ancestral proteins that we tested could have had different effects on the ancestral organism in which they occurred. All transgenic experiments, including horizontal swaps of genes or regulatory elements from one species into a model organism, are potentially complicated by this kind of epistatic mismatch (*Gehrke and Shubin, 2016*). In general, epistatic mismatch should be less of a problem when ancestral genes are used because the genetic distance between ancestral gene and host organism is shorter than it is in horizontal swaps between extant species that descend from the same ancestor. We observed virtually no developmental or regulatory differences between *D. melanogaster* transformed with its own DmBcd-HD and those carrying the AncBcd-HD, suggesting no incompatibility between the ancestral HD and the developmental environment in which it was tested. Moreover, our assays of AncBcd, AncZB, and variants containing Q50K and/or M54R were all carried out in the common genetic background of *D. melanogaster;* so our inferences would be artifacts only if interactions with this background differentially affected these very similar sequences.

A third limitation is that we focused our study on evolutionary change within the Bcd HD, because only the HD can be reconstructed with confidence. Changes elsewhere in the Bcd coding sequence may have made some contribution to the emergence of the protein's new functions. But our experiments establish that changes within the HD are the predominant causes within the Bcd protein of its functional evolution: swapping just the HD of DmBcd-HD into the rest of the Zen protein confers the majority of Bcd-like in vivo functions, including regulation of all major Bcd target genes and rescue of many anterior structures. The level of expression and precise localization of some Bcd targets is not fully rescued by this Zen-BcdHD chimera, however, so amino acid changes outside the HD likely had additional, albeit relatively minor effects, on tuning Bcd's derived gene regulatory functions.

Bcd's evolution involved a shift in not only the protein's intrinsic functional properties but also its localization; our experimental design did not address the genetic changes that drove maternally expressed *bcd* to become localized in the anterior of the embryo. The evolution of Bcd functions may also have involved changes in proteins with which Bcd interacts or in Bcd's target regulatory elements. These questions could, in principle, be addressed in the future using ancestral reconstruction. Indeed, the strategy we use here can be applied to any developmental functions and phenotypes for which phylogenetic signal is adequate to allow relevant ancestral sequences to be reconstructed. Ancestral regulatory regions may be amenable to reconstruction and in vitro/in vivo characterization (*Rebeiz et al., 2011*; *Rogers et al., 2013*), if they are conserved enough that traces of the ancestral state are retained in extant sequences. It may therefore be possible, in some cases at least, to provide more complete causal explanations across biological levels for the ancient evolution of differences in development among distantly related taxa.

## Implications

Our results shed light on several issues concerning developmental evolution. First, they illuminate the number and effect-size of substitutions during the evolution of development, a classic question in evolutionary genetics (*Fisher, 1919*; *Lewontin, 1974*; *Orr, 2005*) and evo-devo (*Colosimo et al., 1928*; *Martin and Orgogozo, 2013*; *Rockman, 2012*; *Stern and Orgogozo, 2008*). We found that two very large-effect substitutions drove the evolution of the majority of the Bcd HD's derived functions. This result is consistent with findings in other protein families that just a few historical substitutions drove evolutionary changes in biochemical specificity in enzymes, transcription factors, and binding proteins (*Siddiq et al., 2017b*). In contrast, previous studies of development that have identified the genetic basis of differences between populations or closely related species have mostly found a more diffuse distribution of effect sizes, with many substitutions of small effect accounting for new phenotypes (*Gompel et al., 2005*; *Frankel et al., 2011*; *Linnen et al., 2013*; *Rebeiz et al., 2009*); however, a major role for large-effect mutations has been observed when the derived phenotype is conferred by loss-of-function mutations (e.g. refs. [*Colosimo et al., 1928*; *Jeong et al., 2006*; *Prud'homme et al., 2006*; *Hoekstra et al., 2006*]).

In the case of Bcd, both large- and small-effect substitutions appear to have been important, with additional contributions by some of the 29 other substitutions on the same branch. These results are not consistent with a model by which evolution occurs through a very large number of changes of extremely small effect or a simple 'hopeful monster' model by which a single mutation is sufficient to confer a novel, high-fitness phenotype. Our findings are more consistent with a model in which a very small number of large-effect substitutions can yield a new but non-optimal function, which is subsequently refined by additional changes of smaller effect (*Orr, 2005*; *Charlesworth and Charlesworth, 1975*).

Our results also have implications for understanding the evolution of Bcd's role in cyclorrhaphan development. The difference we observed between the in vitro and in vivo contributions of the key historical substitutions indicates that the evolution of Bcd's developmental functions cannot be explained entirely by a shift in binding to canonical ZM and BM. Q50K causes the entire shift in preference measured in vitro, but it accounts for a relatively small portion of the HD's in vivo evolution – activation of three of seven major transcriptional target genes, and only a very limited rescue of anterior development; two of these, *hb* and *eve2,* are the same targets that are activated by a construct of the *D. melanogaster bcd* gene containing the *otd* HD, which also contains K50 (*Datta et al., 2018*). Adding M54R to Q50K had no further effect on preference or affinity in vitro, but it conferred a notably more complete rescue in vivo. Residue Arg54 has been proposed to be critical for Cad inhibition (*Niessing et al., 2000*), but the AncZB-Q50K/M54R construct showed no detectable inhibition of Cad in vivo, indicating that M54R's effects are mediated by some other mechanism. Effects on non-canonical, lower-occupancy DNA targets are one possibility (*Datta et al., 2018*; *Ramos and Barolo, 2013*; *Crocker et al., 2015*), as are changes in cooperative interactions with some other transcription factor (*Baker et al., 2012*). When the remaining historical substitutions have been identified that confer inhibition of Cad translation and activation of Bcd's two remaining transcriptional targets (*ems* and *btd*), it should be possible to use the data we present here to disentangle how Bcd's regulatory effects on its various targets interact to specify AP polarity.

A major puzzle is how a new transcription factor could evolve a controlling role in a crucial process of early development, because intermediate evolutionary states that would disrupt AP development would presumably be strongly selected against. It has been suggested that Bcd may have usurped its role as a primary anterior organizing transcription factor from a more ancient protein with Bcd-like specificity, such as Otd, which plays a role in anterior organization in some wasps (*Lynch et al., 2006*). In this scenario, as soon as Bcd's new specificity evolved, it would immediately have taken on a controlling position in the AP gene regulatory network, so long as it was localized to the embryo's anterior. There are other examples of transcription factors taking control of existing integrated regulatory networks during evolution, although the mechanisms by which they did so are not yet known (*Booth et al., 2010*; *Mank and Avise, 2009*; *Matsuda et al., 2002*; *Wallis et al., 2008*).

Our finding that only one or two changes is sufficient to drastically alter Bcd's target specificity both in vitro and in vivo seems to increase the plausibility of the idea that AncBcd could have commandeered control over an ancestral suite of targets that already contained BMs. But a key premise of this model — that Otd or other HD-containing proteins with Bcd-like specificity were the ancestral anterior organizers — is not yet established. In other insect species, including some more closely related to cyclorrhaphans than are wasps and beetles, non-HD transcription factors unrelated to Otd or Bcd are the upstream controlling determinants of AP development (*Klomp et al., 2015*), so it is currently premature to conclude that Otd played this role in the ancestral cyclorrhaphan. Moreover, replacing the HD of *D. melanogaster* Bcd with that of *D. melanogaster* Otd in transgenic flies yields activation of only a few Bcd targets and virtually no rescue of anterior development (*Datta et al., 2018*); this result seem incompatible with the model that Bcd simply commandeered Otd's targets during evolution, unless subsequent evolution of the system has obscured the putative ancestral relationship.

A second premise of this model is that the network of downstream genes that executes AP specification in response to Otd or some other ancestral master anterior factor was already in place in a form similar or identical to that in extant *D. melanogaster*, ready to be commandeered by Bcd when it evolved. Current knowledge of the mechanisms of AP specification in other species is inadequate to establish the long-term stasis of this network. Subsequent changes in the regulatory sequences of some present-day Bcd target genes may have been important in initiating or modifying their

response to Bcd, after Bcd's intrinsic molecular functions were already acquired. Our experiments, which were conducted in the context of the *D. melanogaster* embryo, cannot address this possibility. To infer the likely components of the ancestral network and the relationships among them, it will be necessary to assess the mechanisms of AP specification in numerous non-cyclorrhaphan dipterans, as well as basally branching cyclorrhapans, and evaluate that evidence in a phylogenetic context. Such efforts will refute or corroborate the 'Otd replacement' hypothesis and may suggest other plausible scenarios for how the relatively rapid acquisition by Bcd of new molecular functions allowed it to evolve a controlling role in a crucial integrated process of early animal development.

## Materials and methods

### Drosophila stocks, cloning, and transgenesis

We used the following stocks in these experiments: yw (wild type), ±/*Cyo bcd+;bcd^{E1}/bcd^{E1}*, *Cyo bcd+/Sco;bcd^{E1}/bcd^{E1}*, and ΦC31 (y+);38F1 (w+). We cloned an injection plasmid (piattB40-Bcd) containing two inverted ΦC31-specific recombination sequences (AgeI/HindIII) for Recombination Mediated Cassette Exchange (RMCE), Gmr-GFP (HindIII/AscI), and a polylinker flanked by 1.9 kb *bcd* promoter and 0.8 kb 3'UTR (AgeI/AscI). The *bcd* and *zen* coding regions were amplified by PCR from pBS-SK + cDNA clones, digested with RsrII and AscI and ligated into piattB40-Bcd. We used standard cloning techniques to generate homeodomain swaps and residue changes. Gene Blocks coding for the ancestral HD and variant sequences together with the flanking Bcd coding sequence were obtained from Integrated DNA Technologies (IDT). They were digested with AscI and BspEI and ligated to the piattB40-Bcd vector digested with the same restriction enzymes. All transgenic lines were generated using the ΦC31 integration system, and constructs were integrated into the 38F1 landing site on the second chromosome (*Bateman et al., 2006*). Each transgene was crossed to *Cyo bcd+/Sco; bcd^{E1}/bcd^{E1}* to generate *Cyo bcd+/[transgene]; bcd^{E1}/bcd^{E1}* stocks. Embryos and larvae from homozygous transgenic females were assayed for gene expression and cuticle phenotype.

### In situ hybridization, immunohistochemistry, and image processing

In situ hybridizations were performed as previously described (*Kosman et al., 2004*). Briefly, embryos 1–3 hr AEL (after egg laying) were dechorionated 2 min in bleach and fixed and devitellinized in a biphasic fixation solution containing 3 ml 1X PBS, 1 ml 37% Formaldehyde and 4 ml Heptane for 25 min on a shaker at RT. Fixed and permeabilized embryos were incubated with DIG or Fluorescein-labeled RNA probes and the labeled probes were detected by Alkaline Phosphatase (AP)-conjugated primary antibodies (Cat #11093274910 and #11426338910) by using NBT/BCIP solution. RNA expression was observed by Zeiss Axioskop microscopy. Cuticle preparations were performed on embryos aged 20–24 hr. Larvae were dechorionated for 2 min in bleach, and a 1:1 mixture of methanol and heptane was used to remove the vitelline membrane. Larvae were fixed overnight at 65°C in a 1:4 mixture of glycerol and acetic acid, and mounted in a 1:1 mixture of Hoyer's medium (Anderson, 1954) and lactic acid. Guinea pig anti-Cad (1:400) and Alexa Fluor conjugated 647 donkey anti-guinea pig (1:500, Invitrogen) were used to examine Cad protein expression. All antibodies were diluted in PBT (1X PBS with 0.1% Tween). Data for immunostaining images were collected on a Leica TCS SP5 confocal microscope using the Leica confocal analysis software.

### Phylogenetics and ancestral sequence construction

Protein sequences of Bicoid (Bcd), Zerknullt (Zen) and Hox3 orthologs were collected by BLAST against NCBI databases using Drosophila melanogaster Bcd and Zen HD sequences. Three zen cDNA sequences were reverse transcribed from insect species, *Lonchoptera lutea*, *Platypeza consobrina* and *Rhajio tringarius* (*Stauber, 2001*). Protein sequences were aligned using the multiple sequence alignment program MSAProbs (v0.9.7) (*Liu et al., 2010*). The alignment was manually examined and HD regions were extracted, realigned, and then checked for proper alignment according to known conserved positions. Only one representative sequence was kept if multiple sequence entries showed no sequence diversity among them. The curated HD alignment included 33 HDs from 27 insect species (*Supplementary file 1*). Phylogenetic analysis was performed using PHYML (v3.0) (*Guindon et al., 2010*) and LG model (*Le and Gascuel, 2008*) with gamma-distributed

among-site rate variation and model defined state frequencies, which was the best-fit evolutionary model selected using the Akaike Information criterion implemented in ProtTest software (v2.4) (*Abascal et al., 2005*). Statistical support for each node was evaluated by calculating the approximate likelihood ratio statistic (*Anisimova and Gascuel, 2006*).

Homedomain sequences are short and highly conserved at many sites, so they contain inadequate information to resolve some species relationships by maximum likelihood (*Figure 2—figure supplement 1*). We therefore imposed a tree topology a priori to constrain all species relationships, which are well corroborated from extensive prior research on insect phylogenetics. Branch lengths and model parameters were then optimized on this tree by maximum likelihood. Ancestral HD sequences were reconstructed by the maximum likelihood method using the codeml module of PAML (v4.7) (*Yang, 2007*), assuming the constrained tree, a free four-category gamma distribution of among-site rate variation, and the LG model. A customized Python script was used to extract marginal posterior probability of each amino acid at each site for the nodes of interest from PAML output file (available at https://github.com/JoeThorntonLab/bcd_evolution; copy archived at https://github.com/elifes-ciences-publications/bcd_evolution). We identified all ambiguously reconstructed sites as those at which multiple states have posterior probability >0.20; we synthesized alternative ancestral reconstructions containing the state with the second highest likelihood at all such sites (*Supplementary file 2*). The resulting 'alt-all' ancestors represent the far edge of the cloud of plausible ancestral sequences and allow a conservative experimental test (*Figure 3—figure supplement 1*, *Figure 4—figure supplement 1*, *Supplementary file 3*) of the robustness of functional inferences to statistical uncertainty about the ancestral sequences (*Eick et al., 2017*).

## Protein synthesis and purification

Ancestral HD cDNA sequences were synthesized (GeneScript). For SPR and F-EMSA experiments, HD cDNAs were cloned into the pETMALc-H10T vector (*Pryor and Leiting, 1997*) C-terminal to a cassette containing a 6xHis tag, maltose binding protein (MBP) and a TEV protease cleavage site. For protein binding microarray experiments, cDNA sequences of ancestral HDs plus a flanking region containing 15 amino acids from *D. melanogaster* Bcd or Zen were synthesized (GeneScript) and cloned into the pDEST15 vector through Gateway cloning (Life Technologies) to produce N-terminal GST fusion proteins. Ancestral HDs were expressed in BL21 (DE3) pLysS Rosetta cells.

Protein expression was induced by 0.5 mM IPTG at A600 of 0.8–1.2. Cells were then grown overnight at 18°C, harvested and frozen at −20°C. Cells were lysed using BPER Protein Extraction Reagent Kit (ThermoScientific). For MBP-His fusion ancestral HDs, the supernatant of the cell lysate was loaded onto a pre-equilibrated 5 mL HisTrap HP column (GE) and eluted with a linear imidazole gradient (40 mM to 1 M) in 20 mM sodium phosphate and 200 mM NaCl buffer [pH 7.4] on AKTA FPLC system. Then, ancestral HDs were cleaved from MBP-His fusions using TEV protease in dialysis buffer consisting of 20 mM sodium phosphate and 250 mM NaCl [pH 7.0]. For GST fusion ancestral HDs, the supernatant of cell lysate was loaded onto a pre-equilibrated gravity Glutathione column (ThermoScientific), eluted with 10 mM glutathione, 125 mM Tris and 150 mM NaCl [pH 8.0], and dialysed in dialysis buffer [pH 6.0]. For the second purification step, samples were loaded onto a 5 mL HiPrep SP FF cation exchange column (GE) and eluted with a linear NaCl gradient (250 mM to 1 M) in 20 mM sodium phosphate buffer [pH 7.0]. Whenever needed, ancestral HDs were further purified on a HiPrep 16/60 Sephacryl S-100 HR size exclusion column (GE) with 20 mM Tris and 150 mM NaCl [pH 7.5]. Protein purity was assayed after purification by visualization on a 12% SDS-PAGE gel stained with Bio-Safe Coomassie G-250 stain (Bio-Rad). Purified protein aliquots were flash frozen in liquid nitrogen and stored at −80°C in 20 mM sodium phosphate, 150 mM NaCl, 1 mM DTT and 10% glycerol [pH7.5]. Protein folding and aggregation states were measured using circular dichroism (Jasco J-1500 CD) and dynamic light scattering (Wyatt DynaPro NanoStar) spectroscopy after freeze-thaw cycle. For biochemical assays, protein aliquots were thawed, centrifuged, and the supernatant was used for Quick StartTM Bradford Assay (BioRad) to quantify protein concentrations.

## Protein binding microarray analysis

Protein binding microarrays (PBM) were performed essentially as described previously (*Berger et al., 2006*) using a custom-designed 'all 10-mer' universal array (*Berger et al., 2008*) in the 8 × 60K array format (Agilent Technologies, Inc.; AMADID # 030236) (*Nakagawa et al., 2013*).

All proteins were tested at a concentration of 100 nM. Duplicate PBMs were performed for each protein. The array data were quantified as described previously (*Berger et al., 2006*) using the Universal PBM Analysis Suite (*Berger and Bulyk, 2009*). Replicate experiments for each protein yielded similar results, although the signal in one replicate was in most cases stronger than in the other (*Figure 4—figure supplement 1*, *Supplementary file 6*). For PBM analysis, 15 amino acids flanking the HD were included to follow previously optimized protocols (*Berger et al., 2008*); we found almost identical results whether these residues were those from *D. melanogaster* Bcd or Zen (*Figure 4—figure supplement 1*, *Supplementary file 3*).

For heat-map and clustering analysis, E-scores for 8-mers scoring≥0.45 for at least one tested construct were clustered and plotted using the heatmap.2 function in the gplots R package, with the Manhattan distance metric. Clusters of 8-mers selected from the heatmaps were aligned as described (*Jiang et al., 2013*), and sequence logos were made using the makePWM and seqLogo functions from the Biostrings and seqLogo R packages. For correlation analysis, binding of a HD protein to every 6-mer was calculated as the mean E-score averaged over all 8-mers containing that 6-mer; 6-mers were used because our F-EMSA and SPR experiments were performed using canonical 6-mers, and because the strongest information about sequence specificity observed in our PBMs and those of other homeodomains typically comes from 6-mers, although flanking bases can provide some additional information (see, for example, *Figure 4* of [*Berger et al., 2008*]). The Pearson correlation coefficient was then calculated for the relationship between 6-mer E-scores of PBMs performed using two different HDs.

PBM binding logos were derived from site-specific effects on binding energies estimated using a maximum likelihood method implemented in the BEEML-PBM algorithm (*Zhao and Stormo, 2011*). BEEML-PBM models the nonlinear relationship between the logarithm of binding probability and the binding free energy by including transcription factor concentration as a separate parameter and also taking into account background and position effects. In practice, for each ancestral HD, the normalized probe intensity data and a set of initial parameters specifying its position weight matrix (PWM) energy model were used as input files for nonlinear regression to obtain the parameters of its PWM energy model that maximize the fit to the observed probe intensity data. To evaluate PWM energy model performance, PWMs trained on replicate one were used to predict replicate 2 8-mer median intensities. Observed or predicted median intensity of an 8mer was calculated by taking the median of the observed or predicted intensities of all probes containing this 8-mer. The coefficient of determination ($R^2$) was calculated to quantify the proportion of variance of observed median intensities that could be explained by predicted median intensities. The predictive performance was then compared with experimental reproducibility ($R^2$ of 8mer median intensities) between the two replicates. To evaluate the robustness of our statistical sequence reconstruction, a PWM energy model trained on an alternate reconstructed ancestral HD was used to predict 8-mer median intensities of its corresponding ML ancestor. PWM energy models were graphically represented using energy logos implemented in Energy Model program (*Foat et al., 2006*; *Zhao et al., 2012*), which represent the relative energy contribution for each nucleotide at each position, and directly display binding affinity preference of favorable and unfavorable nucleotides.

## Quantitative fluorescence electrophoretic mobility shift assays (F-EMSA)

Quantitative F-EMSAs were used to measure the binding affinities of ancestral HDs to 14 bp 5' fluorescein-labeled dsDNA oligonucleotides (FAM-DNA). FAM-DNAs were produced by annealing equimolar complementary ssDNA strands (HPLC purified; Eurofins). 5 µM annealed FAM-DNA aliquots were stored at −20°C. Typical reactions consisted of 2.5 nM FAM-DNA equilibrated with varying concentrations of protein in assay buffer for 30 min at room temperature (20 mm Tris, pH 7.5, 150 mm NaCl and 1 mM DTT). Immediately prior to loading, one-twentieth volume of 30% (v/v) glycerol, 0.01% (w/v) bromcresol green was added to each reaction as a dye marker. A 10 µl sample of each reaction was loaded onto a pre-run 8% native polyacrylamide gel in pre-chilled 0.5 × TBE buffer. The gels were run for 40 min at 100 volts then immediately scanned using a fluor-imager (Bio-Rad GelDoc) with a blue laser at 491 nm. The fluorescence intensity ratio of bound verse total FAM-DNA (F) was determined as a function of total protein concentration (P) and DNA concentration (D), and the data were fitted to a quadratic equation (*Equation 1*) to determine the binding association constants ($K_A$). In *Equation 1*, b0 and bM represent the baseline and maximum response as

normalization factor for data fitting. All procedures associated with image processing and model fitting were carried out using a customized R script.

$$F = b_0 + (b_M - b_0) \times \left[ \frac{D + P + K_A^{-1} - \sqrt{\left(D + P + K_A^{-1}\right)^2 - 4PD}}{2D} \right] \quad (1)$$

### Surface plasmon resonance assays

All SPR kinetic experiments were performed with a four-channel Biacore 3000 biosensor system (GE) at 25°C. 20 bp 3′-Biotin-labeled DNA oligonucleotides (HPLC purified; Eurofins) were captured with streptavidin on CM5 sensor chips (GE) on which the carboxymethyl groups were activated and coupled with streptavidin to amine linkages. Flow cell #1 was left blank as a control while others contained a specific immobilized DNA sequence. Preliminary experiments were done to find appropriate buffer and surface regeneration conditions, DNA immobilization levels, protein concentrations, flow rate, association and dissociation contact time for each protein variant. Low levels of DNA immobilization, 100 ~ 150 RUs (response unit), and fast flow rate, 30 µl /min, were used to minimize mass transfer effects. Before each kinetic experiment, the system was stabilized for 3 min by flowing HBS-PE buffer at 30 µl /min (10 mM HEPES pH 7.4, 150 mM NaCl, 3 mM EDTA, 0.005% v/v Surfactant P20). During the kinetic association phase, a protein dilution in modified HBS-PE buffer (10 mM HEPES pH 8.5, 150 mM NaCl, 3 mM EDTA, 0.005% v/v Surfactant P20, 1 mM DTT) was injected. Following a kinetic dissociation phase in which only running buffer was injected, DNA binding surface was regenerated by flowing 20 µl 0.05% SDS at 100 µl /min. The kinetic parameters ($k_{on}$ and $k_{off}$) were estimated by using BIAevaluation software (GE) to fit the observed data (R) to a function (*Equation 2*), with terms for concentration of protein variants ($c_0$) at various time points (t) and correction terms for refractive index change ($R_{RI}$), drifting baseline ($c_D$) and local maximum response ($R_{max}$).

$$R = \frac{k_{on} c_0 R_{max}}{k_{on} c_0 + k_{off}} \left[ 1 - e^{-\left(k_{on} c_0 + k_{off}\right)t} \right] + R_{RI} + c_D t \quad (2)$$

## Acknowledgements

We thank Elena Solomaha, Jackie Moore, Christine Yoo, Anastasia Vedenko, Leila Shokri, and Jesse Kurland for assistance with experiments and analysis. We thank Claude Desplan, Lionel Christiaen, Vincent Lynch, and members of the Thornton and Small laboratories for helpful discussions and comments on the manuscript. Supported by NIH R01GM051946 (SS), R01GM104397 (JWT), R01GM121931 (JWT), R01-HG005287 (MLB), and F32GM112351 (QL), NSF IOS0744966 (SS) and IOS1355057 (US-O) and NYU's Dean for Science Research Fund (SS).

## Additional information

### Funding

| Funder | Grant reference number | Author |
| --- | --- | --- |
| National Institutes of Health | R01GM051946 | Stephen J Small |
| National Institutes of Health | R01GM104397 | Joseph W Thornton |
| National Institutes of Health | R01GM121931 | Joseph W Thornton |
| National Institutes of Health | R01-HG005287 | Martha L Bulyk |
| National Institutes of Health | F32GM112351 | Qinwen Liu |
| National Institutes of Health | NSF IOS0744966 | Stephen J Small |
| National Institutes of Health | IOS1355057 | Urs Schmidt-Ott |
| New York University Dean for Science Research Fund | | Stephen J Small |

The funders had no role in study design, data collection and interpretation, or the decision to submit the work for publication.

## Author contributions
Qinwen Liu, Conceived the project; performed phylogenetic analyses, quantitative gel shifts, and surface plasmon resonance assays; analysed the protein binding microarray experiments data; interpreted data and contributed to writing the manuscript; Pinar Onal, Conceived the project; generated transgenic lines containing HDs from AncBcd, AncZB, AncBcdK50q, AncZBq50K, AncZBm54R, and AncZBK50/R54; performed cuticle preparations, in situ hybridizations, and immunofluorescence assays; interpreted data and contributed to writing the manuscript; Rhea R Datta, Conceived the project; generated transgenic lines containing HDs from DmBcd, DmZen, and DmZenBcdHD; performed cuticle preparations, in situ hybridizations, and immunofluorescence assays; analysed the protein binding microarray experiments data; interpreted data and contributed to writing the manuscript; Julia M Rogers, Performed protein binding microarray experiments; analysed the protein binding microarray experiments data; interpreted data and contributed to writing the manuscript; Urs Schmidt-Ott, Conceived the project; interpreted data and contributed to writing the manuscript; Martha L Bulyk, Contributed to the design of experiments and analyses; interpreted data and contributed to writing the manuscript; Stephen Small, Conceived and supervised the project; contributed to design of experiments and analyses; interpreted data and contributed to writing the manuscript; Joseph W Thornton, Conceived and supervised the project; contributed to the design of experiments and analyses; interpreted data and contributed to writing the manuscript

## Author ORCIDs
Urs Schmidt-Ott (iD) https://orcid.org/0000-0002-1351-9472
Joseph W Thornton (iD) https://orcid.org/0000-0001-9589-6994

## Decision letter and Author response
Decision letter https://doi.org/10.7554/eLife.34594.022
Author response https://doi.org/10.7554/eLife.34594.023

## Additional files

### Supplementary files
• Supplementary file 1. Extant homeodomain sequences used for phylogenetic analysis and ancestral reconstruction. For each sequence in the alignment, the genus and species name, amino acid sequence, and abbreviation used in the phylogeny are shown.
DOI: https://doi.org/10.7554/eLife.34594.012

• Supplementary file 2. Reconstructed ancestral sequences and site-specific marginal posterior probabilities (mpp). At each sequence site, the three amino acid states with the highest mpp are shown for the HDs of AncZB (panel A) and AncBcd (panel B). Red indicates ambiguously reconstructed sites, defined as those that have more than one state with mpp >0.20.
DOI: https://doi.org/10.7554/eLife.34594.013

• Supplementary file 3. PBM binding specificity profiles are robust to the uncertainty in ancestral sequence and choice of flanking sequences. Energy PWMs were inferred from PBMs using maximum likelihood reconstructions (no suffix) or alternative reconstructions (suffix 'altAll') and from constructs using 15 flanking residues derived from *D. melanogaster* Bcd protein (prefix 'B-') or from *D. melanogaster* Zen protein (no prefix). The Predicted column shows $R^2$ for the comparison of intensities predicted from a model fit to the PBM in the column labeled 'Training' against measured intensities in the PBM of the 'Prediction' protein. The 'Measured' column shows $R^2$ for comparison of measured intensities from the training PBM against the prediction sample.
DOI: https://doi.org/10.7554/eLife.34594.014

• Supplementary file 4. Kinetic binding parameters inferred by surface plasmon resonance assays. Binding by various homeodomain constructs (HD) to canonical DNA motifs BM or ZM was measured by SPR. The estimated on-rate ($k_{on}$), off-rate ($k_{off}$), association constant ($K_A$) and average residence time $t_{1/2}$ are shown. Mean and SEM of three replicates is shown for each parameter.
DOI: https://doi.org/10.7554/eLife.34594.015

• Supplementary file 5. Frequency of phenotypes in transgenic embryos. Frequency of observation (in percent) of various phenotypic features in cuticular preps of embryos from females of various genotypes is shown. YW, wild-type; *bcd-*, *bcd^E1^/bcd^E1^*; AncZBq50K, *bcd-* embryos rescued with a *bcd* construct containing the ancestral AncZB homeodomain with the q50K substitution; AncZB-K50R54, *bcd-* embryos rescued with a *bcd* construct containing the ancestral AncZB homeodomain with the q50K and m54R substitutions. No mutant larvae showed complete head development; partial head is defined as the observation of microscopic bright spots (sclerotized tissue) at or near the anterior tip. T1-T3: Thoracic segments, numbered anterior to posterior, A1-A4: Abdominal segments, numbered anterior to posterior. The number of larvae counted for each genotype, *n*, is shown.
DOI: https://doi.org/10.7554/eLife.34594.016

• Supplementary file 6. Protein binding microarray experimental reproducibility and binding model fitting results. The table shows model accuracy and cross-replicate reproducibility for protein binding microarrays. Each row represents one HD tested in two replicates. Energy models (see Materials and methods) were trained on data from replicate 1, used to predict the median intensity of each possible 6-mer, and compared to the experimentally measured median intensity of DNA probes carrying that 6-mer (see Materials and methods). Column a, coefficient of determination ($R^2$) for model-predicted intensities from PBM replicate 1 against measured intensities from replicate 1. Column b, $R^2$ for model-predicted intensities from replicate 2 against measured intensities for replicate 2. Column c, $R^2$ for measured intensities from PBM replicate 1 against measured intensities from PBM replicate 2. Cross-replicate statistical reproducibility is reduced because stripping and re-using protein binding microarrays reduces signal; nevertheless, the specific binding profiles are similar between replicates (see *Supplementary file 6*).
DOI: https://doi.org/10.7554/eLife.34594.017

• Transparent reporting form
DOI: https://doi.org/10.7554/eLife.34594.018

## Data availability

The PBM data generated is available via Dryad (doi:10.5061/dryad.pm3g4r3)

The following dataset was generated:

| Author(s) | Year | Dataset title | Dataset URL | Database, license, and accessibility information |
|---|---|---|---|---|
| Liu Q, Onal P, Datta R, Rogers J, Schmidt-Ott U, Bulyk M, Small S, Thornton J | 2018 | Data from: Ancient mechanisms for the evolution of the Bicoid homeodomain's function in fly development | http://dx.doi.org/10.5061/dryad.pm3g4r3 | Available at Dryad Digital Repository under a CC0 Public Domain Dedication |

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
