## [Decision Letter]

Thank you for submitting your work entitled "Ancient mechanism for the evolution of Bicoid function in fly development" for consideration by *eLife*. Your article has been reviewed by three peer reviewers, and the evaluation has been overseen by a Senior Editor. The following individuals involved in review of your submission have agreed to reveal their identity: Artyom Kopp (Reviewer #3). The other two reviewers remain anonymous.

Our decision has been reached after consultation between the reviewers. Based on these discussions and the individual reviews below, we regret to inform you that your work will not be considered further for publication in *eLife*.

All three reviewers agree that the methodology is a significant strength of the paper. After discussion, however, the reviewers and editors felt that the results presented here are very much in line with what is already known about the functional changes brought about by evolutionary amino acid substitutions in the BCD homeodomain. Furthermore, the conclusions were based on testing variants in *Drosophila melanogaster*, which does not allow for an analysis of the contributions of "trans" evolutionary changes in target sequences, and in interacting proteins. In the end, it was the lack of a significantly new advance in understanding BCD protein evolution/function that led to the decision.

*Reviewer #1:*

This study sought to characterize the amino acids changes that have evolved to confer functionality on Bicoid since the ancestral duplication that produced this gene in the lineage leading to cyclorrhaphan flies. The study focused on the homeodomain and used a transgenic rescue assay to test the effect of variants on target gene expression and overall ability to rescue in vivoin vivo. These experiments were complimented by in vitroin vitro analysis of binding site preferences of variants.

The authors found that the Bcd homeodomain in an otherwise Zen protein could perform much like Bcd. They then used ancestral sequence reconstruction (ASR) to generate and test ancestral sequences. This allowed them to identify an ancestral-like sequence that could function like Bcd although it was unable to fully function like this transcription factor, in for example, repressing translation of caudal. The authors then sought to determine which Bcd specific amino acids in the homeodomain were responsible. They found that the K at position 50 conferred much of the characteristics of Bcd in vitroin vitro and in vivoin vivo and that the R at position 54 also contributed to Bcd function in vivoin vivo. None of the other amino acid differences tested made any detectable contribution, although since rescue was not complete they and other changes outside the homeodomain presumably also have minor roles.

Overall the experiments are described clearly and the work confirms that the amino acids at positions 50 and 54 have made a major contribution to the evolution of Bcd functionality as inferred from several previous studies cited in the manuscript.

A case is made in the Discussion sectionDiscussion section for how these amino acid changes may have facilitated the usurpation of the ancestral gene regulatory network for anterior patterning. However this study only assesses changes in the homeodomain of Bcd and therefore there appears to be an assumption that the binding sites in the target genes have not changed in sequence and nor has there been any other changes in trans. Therefore this speculation should be toned down or at least the caveats should be explained in more detail.

It would also have been interesting to determine how Otd with a Bcd homeodomain and other variants with particular amino acid substitutions performed in these assays to further test what ancestral system Bcd replaced and how this was achieved.

The Introduction sets out the case for the merits of the ASR approach to identify mutations underlying evolutionary differences. However, while ASR can be a powerful approach, I think the Introduction is too biased and there is an implication that horizontal comparisons do not generally allow the genetic changes underlying phenotypic differences to be identified (e.g. from wording such as 'differences that correlate with phenotypic differences' and 'identified allelic differences that reproduce diversity in gene expression and developmental outcomes'. On the contrary mapping approaches and functional analyses of variants between species have identified many mutations underlying population and species differences (e.g. see Martin and Orgogozo, 2013) and continue to do so. Therefore a more balanced Introduction is needed.

*Reviewer #2:*

The article by Liu et al.et al. aims to understand the evolution of the very interesting fly protein Bicoid. They use a cutting edge tool for reconstructing the most likely ancestor of it and its paralog Zen, and testing the functions of variations on the modern and ancestral proteins in establishing and patterning the anterior posterior axis. I think the work is very well done, but in the end, does not advance our knowledge of the properties and evolution of Bicoid significantly. The importance of the transition from Q50 to K50 is well known and has been for around 30 years, and it is no surprise that it is far and away the most significant change in the protein. Position 54 is also well known to have important functions.

These results should be the starting point of the paper, not the main conclusions. The really interesting thing to me is why these are not sufficient to give complete Bicoid rescue? Is it still a question of intramolecular epistasis that prevents the combination of k50 and r54 from activating all bicoid targets? If so, which sites are involved? Or is something else going on? Perhaps the cis-regulatory elements in D*., melanogaster* have coevolved with the *Drosophila* Bcd homeodomain? I realize testing the latter hypothesis is outside the scope of the approach of this paper, but a more systematic probing of the potential of interactions among residues within the homeodomain with the two large effect changes could support or eliminate the intramolecular epistasis hypothesis, and could give more support to searching for other explanations.

As it stands I feel that the field would not be advanced significantly by this work, since we are still at the point of knowing that q50 to k50 led to a big change in the sequences bound by Bicoid, but we have no further understanding of how this led to Bicoid becoming the anterior determinant of the cyclorrhaphans.

*Reviewer #3:*

I think this is an exemplary paper. As the authors explain in their very eloquent Introduction, incorporating explicit phylogenetic structure and ancestral character reconstruction into comparative studies produces much more rigorous results than simple pairwise comparisons between species. This is as true for development as it is for morphology and DNA sequences, but unfortunately the application of phylogenetic approaches is much rarer in evo-devo studies than in other sub-fields of evolutionary biology – – due at least in part to the difficulty of collecting appropriate data, but also perhaps to cultural factors. This paper raises the bar in both respects. The authors have performed a very nicely structured comparison of reconstructed ancestral proteins to investigate how their function evolved through time on a defined evolutionary lineage. At the same time, they have integrated a wealth of in vivoin vivo, in vitro, and biophysical data to characterize the functions of both extant and ancestral proteins. The result is a very thorough and rigorous paper. In some respects, the results are not entirely surprising – – the single mutation that was found to play by far the largest role in the evolution of protein function is the one that has been hypothesized to do just that for decades. However, in addition to confirming the role of this mutation, the authors were able to quantify its contribution to phenotypic change at different levels of biological organization (from binding affinity/occupancy rate to target gene expression to morphological phenotype), identify a second AA substitution that acts in concert with it, and to rule out the role of multiple other fixed AA changes (within the limits of their assays, of course). So the exceptionally laborious approach taken by the authors has paid off.

[The decision letter after re-review follows.]

As you know, the initial reviews and discussion identified this as interesting and unusually rigorous work demonstrating the phenotypic effects of lineage specific amino acid substitutions in flies. However, concerns were also expressed about the novelty of this work and impact on the field given that the amino acids tested had previously been suggested to be involved in the trait studied. After much debate, I decided initially to reject this work based on these grounds. You have done a great job highlighting the differences between this prior work and your new findings in the comprehensive response to reviewers and appeal letter. After consulting with all three of the original reviewers, we agree that the changes to the manuscript address our concerns. I am thus happy to now accept this work for publication.

---

## [Author Response]

[Editors’ note: the author responses to the first round of peer review follow.]

Thank you for the decision letter and reviews of our paper. As the letter states, all reviewers viewed the paper’s methodology as very strong, and there were no concerns about the quality of the experiments, the data, or their interpretation. The major comments pertained to the paper’s impact in relation to prior work and, to a lesser extent, to the way we framed a few issues in the introduction and discussion. We have revised the paper to address these concerns, as detailed below.

1) Reviewers 1 and 2 commented that the impact of the paper was low, because the importance of sites 50 and 54 was already known. We believe this comment arises because we inadequately described previous work in our relation to our findings, because a close reading of the literature shows that the extent to which Q50K and M54R can account for the evolution of Bcd’s derived functions in vitro was not previously known, and in vivo, there was no evidence whatsoever to indicate that these substitutions can confer Bcd-like functions or played any role in Bcd evolution.

As the reviewers pointed out, numerous previous studies showed that swapping lysine and glutamine at site 50 between several HD proteins strongly affects in vitro DNA preference for Bcd vs. Zen motifs, and other experiments studied the in vitro effects of various mutations at site 54. Thus, it was reasonable to hypothesize that these might have been involved Bcd evolution. However, these studies could not and did not quantify the extent to which the evolutionary acquisition of Bcd’s derived specificity was caused by the specific historical sequence changes that occurred at sites 50 and 54. That goal would require quantifying the historical shift in Bcd affinity for both kinds of target sequence and assessing the effects of the historical substitutions, in the ancestral background in which they occurred; none of this was previously done. in vivo, no studies have provided any evidence that these substitutions caused the evolution of Bcd’s role in gene regulation and development: the only relevant work found that introducing Q50K into a related HD confers *none* of Bcd’s derived functions, and the M54R substitution’s effects on Bcd functions was never previously assessed.

Our findings are therefore unprecedented that (1) Q50K alone can confer significant (but partial) in vivo rescue of anterior gene regulation and development, (2) adding M54R yields a substantially more complete rescue of these phenotypes, (3) M54’s effect is epistatically dependent on co-introduction of Q50K, (4) Other substitutions from the same phylogenetic interval are required for complete acquisition of all of the Bcd HD’s regulatory activity and role in anterior development, and (5) the acquisition of Bcd’s derived DNA specificity in vitro is entirely attributable to Q50K.

We have revised the Introduction and Discussion section to thoroughly discuss prior work and to explicitly place our strategy and the impact of our findings in the context of that work.

2) The decision letter and reviewer 2 expressed concern that our in vivo experiments were conducted by introducing variants of the ancestral Bcd into *D. melanogaster;* this design, the comment says, did not allow us to test the evolutionary consequences of changes at Bcd’s regulatory targets or interacting proteins, which could also have been important in the evolution of its role in AP development. It is true that we focused on the evolutionary effect of changes in the HD, using *D. melanogaster* as a transgenic platform, but we believe this design is a strength of our study, so long as the interpretations are drawn appropriately. All transgenic experiments are tests of specific putative causal factors when introduced into a particular genetic background; they cannot address the contributions of other potential causes that are already “embedded” within the host organism, and they must be interpreted accordingly. In our study, changes at other loci could indeed have played a role in the evolution of Bcd’s developmental role, but our experiments quite decisively establish the developmental effects of the genetic changes within the Bcd HD that we tested. Further work will be necessary to study the impacts of changes at other loci.

A second potential concern could relate to the possibility of an epistatic mismatch between the ancestral gene and the host organism into which it is introduced, potentially leading to artifactual results. This issue, which potentially affects all transgenic studies, is important to acknowledge. But a strength of our ancestral reconstruction strategy is that it can help to reduce this potential problem compared to horizontal transgenic experiments, because an ancestor and descendant differ only by the changes that occurred along only one branch, whereas the two extant species differ by all changes that occurred along both branches. Further, it is unlikely that our particular findings are attributable to epistasis between ancestral alleles and the transgenic host in which they were assayed, because the developmental and regulatory phenotypes of *D. melanogaster* transformed with its own DmBcd-HD and those transformed with the AncBcd-HD are essentially indistinguishable. Finally, our inferences involved tightly focused comparisons, in which we tested ancestral proteins from successive phylogenetic nodes and examined the effect of specific sequence changes on the phenotypes they produce; this design drastically limits the scope of the inferences that might plausibly be compromised by epistasis with the assay system, because the proteins we tested are so similar to each other. Only if epistasis differentially affects the specific substitutions Q50K, M54R, and the other differences between AncZB and AncBcd *–*which were assessed in otherwise identical proteins in a common genetic background *–*could our results be misleading.

We have modified the DDiscussion section and Iintroduction to explicitly acknowledge the potential importance of evolutionary change at Bcd targets at interacting proteins and to carefully describe the issue that arise when ancestral genes are assessed by introducing them into an extant model organism.

3) Reviewer 1 felt that our introduction was too dismissive of comparative strategies for identifying the genetic basis for the evolution of development. We agree that such approaches have been very important, particularly among closely related species among populations, and would not want imply otherwise. Among more distant taxa, they have also made significant contributions to our understanding, although typically at considerably lower resolution. We have modified the Introduction to acknowledge these contributions.

4) Reviewer 1 suggested that we remove the phrase “higher taxa.” We have replaced it with “more distantly related taxa.”

5) Reviewer 1 suggested that we refer to the Bcd motif using a longer sequence than the canonical 6-bp core sequence we described. We prefer the classic 6-bp core because it contains essentially all of the key information about Bcd’s mode of DNA recognition relative to Zen, which is the relevant change on the evolutionary timescale studied here. Further, it is consonant with our PBM and EMSA experiments, and it is far more accessible for the reader than a longer motif that includes a small amount of degenerate information at flanking sites. We have provided appropriate citations for the consensus sequence we use and have modified the text to acknowledge that there is some additional sequence preference at flanking sites.

6) Reviewer 1 asked us to fix mismatches in the references and check all figure pointers; we did so.

7) Reviewer 2 and reviewer 3 found aspects of our introductory description of the relationships among Hox3, Zen, and Bcd to be confusing. We have rewritten this section for accuracy and clarity.

8) Reviewer 2 suggested we change description of the evolutionary relationship between Bcd and Ftz. We have done so, referring to Ftz as a distantly-related HD protein rather than a paralog.

9) Reviewer 3 asked for more information about the constrained topology and why we used it. We have modified the methods section to more fully address this. Briefly, HD sequences are too short and have inadequate phylogenetic signal to resolve relationships among all the species with sequences in our alignment. These species relationships, however, are all well established by previous work using much larger datasets. We therefore imposed these relationships in a single tree topology, and then used maximum likelihood to optimize branch lengths and model parameters and infer ancestral sequences. We have explained this in the text.

10) Reviewer 3 asked if including the Syrphid sequence affects the ancestral sequence. We checked, and the effect is trivial: reconstructions of AncBcd with or without EbaBcdHD have the same ML sequence, and the difference in their marginal average posterior probabilities is 0.006.

11) Reviewer 3 asked about the correlation of DNA occupancy in PBM between Dn-Zen and AncZB. We cannot answer that question, because we did not perform a PBM for DmZen.